# Direct U-Pb dating of carbonates from micron scale fsLA-ICP-MS images using robust regression

Guilhem Hoareau[1], Fanny Claverie[2], Christophe Pecheyran[2], Christian Paroissin[3], Pierre-Alexandre Grignard[1], Geoffrey Motte[1], Olivier Chailan[4] and Jean-Pierre Girard[4]

[1] Universite de Pau et des Pays de l'Adour, E2S UPPA, CNRS, TOTAL, LFCR, UMR5150, 64000 Pau, France
[2] Universite de Pau et des Pays de l'Adour, E2S UPPA, CNRS, IPREM, UMR5254, 64000 Pau, France
[3] Universite de Pau et des Pays de l'Adour, E2S UPPA, CNRS, LMAP, UMR5142, 64000 Pau, France
[4] TOTAL, CSTJF, F-64018 Pau Cedex, France

*Correspondence to*: Guilhem Hoareau (guilhem.hoareau@univ-pau.fr), Christophe Pécheyran (christophe.pecheyran@univ-pau.fr)

**Abstract.** U-Pb dating of carbonates by LA-ICP-MS spot analysis is an increasingly used method in the field of geosciences, as it brings very strong constraints over the geological history of basins, faults or reservoirs. Most ages currently published are based on the measurement of U and Pb ratios on spot ablations, using nanosecond lasers coupled to sector-field or multi-collector ICP-MS. Here, we test a new strategy for the U-Pb dating of carbonates from 2D isotopic ratio maps, based on the use of a robust regression approach in the data reduction workflow. The isotopic maps, having minimum area of 0.65 mm$^2$ (~1000 pixels of 13x25 μm resolution), are obtained using a 257 nm femtosecond laser ablation system at high repetition rate (500 Hz) coupled to a High Resolution ICP-MS. The maps commonly show significant variations in isotope ratios at the pixel scale, allowing the plotting of pixel U-Pb ratios in concordia or isochron diagrams and the calculation of U-Pb ages. Due to the absence of individual ratio uncertainties, the ages are calculated by using MM-robust linear regression rather than the more commonly used York-type regression. The goodness-of-fit to the data is assessed by the calculation of the residual standard error of the regression (RSE), and by the calculation of a MSWD on discretized data. Several examples are provided that compare the ages calculated by robust regression with those obtained by other techniques (isotope dilution, LA-ICP-MS spot analyses, pixel-pooling approach). For most samples, characterized by high U concentrations (> 1 ppm), robust regression allows calculating ages and uncertainties similar to those obtained with the other approaches. However, for samples with lower U concentrations (< 0.5 ppm), the ages obtained are up to 10% too young due to pixels with high U/Pb acting as leverage points for the regression. We conclude that the U-Pb ages calculated by the regression method tested here, although being statistically robust, should be critically analysed before validation, especially for samples with low U concentrations.

# 1 Introduction

Since the seminal publication of Li et al. (2014), the U-Pb dating of carbonates by LA-ICP-MS has progressed significantly with applications as varied as the dating of tectonic fractures and veins (e.g., Roberts & Walker, 2016; Beaudoin et al., 2018; Parrish et al., 2018; Nuriel et al., 2019), carbonate deposition (Drost et al., 2018), speleothems (Woodhead et al., 2019) or cements filling the porosity of reservoirs (e.g., Li et al., 2014; Godeau et al., 2018). This is due, among other benefits, to the very good spatial resolution of the method, which allows to detect and exploit sub-millimetre scale heterogeneities in U and

initial Pb concentration, to date several samples per day, as well as to the recent availability of carbonate standards to correct matrix-related offsets of U/Pb ratios (Roberts et al., 2017). Given the ubiquitous nature of carbonates, the technique is likely to be associated with a growing number of studies in the future (Roberts et al., 2019).

Most carbonate dating LA-ICP-MS studies are based on spot ablations of sizes ranging from 80 to 235 μm (e.g., Roberts and Walker, 2016; Parrish et al., 2018; Ring and Gerdes, 2016), where isochrons can be constructed from the combination of

40 several tens of ablation craters made on the same crystal or, by default, on adjacent crystals when their size is too small (e.g. micrite). There is now enough hindsight on the applicability of the method to show that this approach allows precise ages to be obtained (< 3% in the most favourable cases) (Roberts et al., 2019). In addition, the fact that most ablation cells can accommodate multiple samples simultaneously makes it possible to automatically analyze multiple samples per session, including primary and validation reference materials (RMs). As recently recalled by Roberts et al. (2019), the heterogeneous

nature of carbonates requires, however, prior petrographic and preferably geochemical characterization of the samples to maximize the chances of successful dating. An initial screening of samples by LA-ICP-MS to produce images of diagnostic trace element concentrations is particularly adapted to localize U-rich and unaltered zones, in which precise and meaningful ages can be obtained (Roberts et al., 2019).

An alternative carbonate dating method based on age extraction from LA-ICP-MS maps was recently presented by Drost et al.

(2018). As before, the simultaneous analysis of many trace elements (including of course U and Pb isotopes) allows visualizing the most favourable areas for dating. However, in this case, the concentrations of elements characteristic of detrital contamination (e.g., Zr, Rb etc), or that allow to define alteration zones or distinct calcite cement generations (e.g., U, Pb, Fe, Mn), are used to define pixel exclusion thresholds. These are used to filter out the values corresponding to microscopic inclusions of other minerals (e.g., clays) and to altered zones, and/or to isolate the cement generation targeted for dating. The

isotopic U and Pb values of the remaining pixels are used to build isotopic ratio maps, from which a U-Pb age can be calculated. The methodology of age determination proposed by these authors is based on the sorting of pixel ratio values used in dating ($^{238}U/^{206}Pb$, $^{207}Pb/^{206}Pb$ for a Tera-Wasserburg (TW) diagram) based on $^{238}U/^{208}Pb$ and $^{235}U/^{207}Pb$ pixel ratios and their pooling (i.e., discretization) to subsets ('pseudo-analyses') of similar number of pixels for which the mean values and associated standard errors (i.e., standard deviations of the mean) are calculated. These values are then plotted in TW, 86TW

($^{208}Pb_{common}/^{206}Pb$ versus $^{238}U/^{206}Pb$; Parrish et al., 2018) and isochron diagrams ($^{206}Pb/^{208}Pb_{common}$ versus $^{238}U/^{208}Pb_{common}$) to calculate the age using York-type regressions as implemented in Isoplot (Ludwig, 2012) or IsoplotR (Vermeesch, 2018).

In this contribution, we test the pros and cons of an alternative approach for the calculation of U-Pb carbonate ages from isotopic maps. Rather than the pooling of pixel values proposed by Drost et al. (2018), the pixel isotopic ratios of interest are directly plotted in the concordia or isochron diagrams, and a statistically robust linear regression that is extremely robust to outliers is processed through the pixels to obtain the age. The visualization of all pixel ratio values in the TW or isochron diagrams allows the user to assess visually the quality of data, guided by a more common statistical parameter such as the residual standard error (RSE). Discretization is also used to calculate pseudo-MSWD values that permit assessing the linearity of pixel data. The robust regression method appears to be ideally suited to isotopic maps obtained after pixel filtering based on trace element concentrations, as proposed by Drost et al. (2018), or if this is not feasible, using pixel colocalization, as proposed here.

## 2 Method

### 2.1 Choice of samples

Six samples of already dated lacustrine and reefal carbonates, and of tectonic veins, have been chosen to test the usefulness of the robust regression approach. Among these, four samples were analysed by the LA-ICP-MS isotope mapping approach as part of this study. These samples are:

(i) a lacustrine limestone (Long Point; Duff Brown Tank locality in the Colorado Plateau, USA), precisely dated by Hill et al. (2016) at $64.04 \pm 0.67$ Ma (2s) by U-Pb methods using isotope dilution (ID) MC-ICP-MS, and labelled Duff Brown in the following. This sample serves as a validation RM in the analytical procedure and has also been demonstrated to be suitable for U-Pb dating using image maps by Drost et al. (2018);

(ii) a tectonic calcite vein (BH14) from the Bighorn basin (Wyoming, USA) dated by U-Pb LA-ICP-MS spot analyses (i.e., range of spot ablations) at $63.0 \pm 2.2$ Ma (MSWD = 1.6) by Beaudoin et al. (2018) and at $61.2 \pm 2.9$ Ma (MSWD = 4.1) in our laboratory (the detailed methodology is presented in Figure S1), with the same calcite RMs as the present study (WC1 and Duff Brown);

(iii) a reefal limestone sample from the southern Pyrenees (PXG20-1), located in the upper part of the Pamplona marls in the Atarés anticline. This sample has been dated at $40.5 \pm 2.3$ Ma (MSWD = 0.9) in our laboratory by LA-ICP-MS spot analysis. A micropaleontologic study from Canudo and Molina (1988) performed close to the sampling area suggests an upper Bartonian to Priabonian age for the deposits (~38.5-36 Ma), whereas the paleomagnetic study of Oms et al. (2003) are more consistent with a Bartonian age (~40-38.5 Ma);

(iv) a lacustrine carbonate from the southern Pyrenees (PXG32-2), which is part of the first continental deposits of the Campodarbe Formation flanking the western limb of the Boltana anticline (eastern Jaca basin). Previous magnetostratigraphic and tectono-sedimentary studies point to a Bartonian-Priabonian age of deposition (Mochales et al., 2016). A similar age was obtained by U-Pb LA-ICP-MS spot analysis in our laboratory, although with a poor statistics ($37 \pm 3$ Ma; MSWD = 6.3)

The two remaining samples were previously analysed by Drost et al. (2018) and Roberts et al. (2019). The raw pixel
isotopic ratios retrieved from their LA-ICP-MS isotopic maps (K. Drost, personal communication) were processed
with the robust regression approach. These are:

(v) a calcite spar from a Carboniferous-Permian lacustrine limestone (sample JS4, run JS4-1), dated by Drost et al. (2018)
at $300.8 \pm 3.7$ Ma, $299.6 \pm 4.0$ Ma, and $298.9 \pm 5.6$ Ma with their pixel pooling approach (TW, 86TW and isochron
diagrams, respectively). This age agrees with several depositional age constraints (see Drost et al., 2018 for details).

(vi) a tectonic vein from the Bighorn basin (Wyoming, USA) dated by U-Pb LA-ICP-MS spot analyses at $59.5 \pm 2.7$ Ma
(MSWD = 2.5) by Beaudoin et al. (2018). Roberts et al. (2019) obtained a similar age of $61.0 \pm 1.7$ Ma with the pixel
pooling approach of Drost et al. (2018).

Finally, three additional, undated samples of calcite cements found in tectonic breccias and veins affecting Tithonian
limestones of the northern Pyrenees (France) are also presented (ETC2, ARB and C6-265-D5). They provide examples of
application of the technique to the resolution of the diagenetic evolution of areas with complex tectonic evolution.

## 2.2. Analytical strategy

### 2.2.1. LA-ICP-MS instrumentation

All the samples were analyzed with a femtosecond laser ablation system (Lambda3, Nexeya, Bordeaux, France) coupled to an
SF-ICP-MS Element XR (ThermoFisher Scientific, Bremen, Germany) fitted with the Jet Interface. The laser is fitted with a
diode-pumped Yb:KGW crystal laser source (HP2, Amplitude Systèmes, Pessac, France). The pulse duration is less than 400
fs at 257 nm. The laser source can operate within a wide range of repetition rates (1 Hz to 100 kHz) and energy ranging from
200 µJ per pulse below 1 kHz to 1 µJ at 100 kHz at 257 nm. Complex trajectories can be realized by moving the laser beam
(15 µm diameter at full energy) across the surface of the sample using the rapid movement of galvanometric scanners combined
with a high repetition rate (Ballihaut et al., 2007, Aramendia et al., 2015) (Fig. 1A).

The aerosol produced by the ablation was carried to the ICP-MS by a tube (1/16" internal diameter) using a Helium stream
($600$ mL min$^{-1}$, with exception for the first session in October 2018, where argon was used ($650$-$700$ mL min$^{-1}$)). Measured
wash out time of the ablation cell was $\sim 500$ ms for helium gas and $\sim 1$ s for argon gas considering the 99% criterion. To
improve sensitivity, 10 mL min$^{-1}$ of nitrogen was added to the twister spray chamber of the ICP-MS via a tangential inlet while
helium flow was introduced via another tangential inlet located at the very top of the spray chamber. Measurements were
performed under dry plasma conditions. The fs-LA-ICP-MS coupling was tuned on a daily basis in order to achieve the best
compromise in terms of sensitivity, accuracy, particles atomization efficiency and stability. The additional Ar carrier gas flow
rate, torch position and power were adjusted so that the U/Th ratio was close to 1 +/- 0.05 when ablating the NIST SRM612
glass. Detector cross-calibration and mass bias calibration were checked daily using the appropriate sequence of the Element
Software. The laser and ICP-MS parameters used for U-Pb dating are detailed in Annex A. For sample screening (first type of

isotopic map, see section 2.2.2), only $^{238}$U, $^{207}$Pb, and $^{206}$Pb were selected, reaching a total mass sweep times of about ~500 ms. For the maps used for dating, $^{238}$U, $^{232}$Th, $^{208}$Pb, $^{207}$Pb, and $^{206}$Pb were selected, reaching a total mass sweep times of about ~60 ms. The analysis of trace elements with lower atomic masses, in a purpose of pixel selection as done by Drost et al. (2018) was not attempted due to the inability of the sector-field ICP-MS to achieve reasonable total mass sweep times (due to magnet mass change). Based on NIST612, the limit of detection as of October 2019 was ~1.3 ppb and ~0.2 ppb for $^{206}$Pb and $^{238}$U, respectively. The limit of quantification was ~4.3 ppb and ~0.8 ppb for $^{206}$Pb and $^{238}$U, respectively.

### 2.2.2. Analytical workflow

Two types of isotopic maps were produced:

(i) The first one (screening map) was designed to identify the most favourable areas for dating (high U/Pb), over a large surface of the sample. In this configuration, samples were first ablated along lines of 6.5 to 8.1 mm width at a repetition rate of 300 Hz, without pre-cleaning. The lines of 50 μm height were obtained using a back and forth movement of the laser (at 10 mm.s$^{-1}$) combined with a stage movement rate of 100 μm.s$^{-1}$ (Fig. 1A), corresponding to 64.5 to 81 s of analysis per line, followed by 15 seconds of break between the lines (to allow ICP-MS data processing). The lines were separated by a distance of 100 μm, leaving a gap of 50 μm between them. The number of lines was of 40 to 56, resulting in a duration of 53 to 78 minutes for a complete map of surface comprised between 26 and 38.1 mm$^2$. The maps produced ($^{238}$U/$^{206}$Pb) are presented in Figure S2.

(ii) The second one (dating map) was designed to date the samples. Therefore, they were performed in selected area (from screening map, and after being repolished) and consisted in linear scans of 0.8 to 1.6 mm width at a repetition rate of 500 Hz. These lines of 25 μm height, separated by a distance of 25 μm so that they are immediately adjacent to each other, were obtained using a back and forth movement of the laser (at 5mm.s$^{-1}$) combined with a stage movement rate of 25 μm.s$^{-1}$, corresponding to 32 to 64 s of analysis per linear scan, followed by 15 seconds of break. The number of lines was of 25 to 27, resulting in a total analysis time ranging from 19 to 38 minutes for a complete map of surface comprised between 0.5 and 1.1 mm$^2$ (Fig. 1B). Before analysis, the samples were pre-cleaned with the laser using a stage movement rate of 200 μm.s$^{-1}$. The unknowns were bracketed with the NIST SRM612 for lead ratios, and by the commonly used WC1 calcite RM (Age 254.4 ± 6.4 Ma; Roberts et al., 2017) for the Pb/U ratio, using the standardization method of Roberts et al. (2017). Two unknowns could be analysed in a row between the RMs. The primary and validation RMs were analysed in conditions similar to the unknowns, except that the isotopic maps were of smaller surface (~0.2 mm$^2$), corresponding to an analysis time of ~5 minutes. The ablation depth was comprised between ~25 μm and ~35 μm.

## 2.3. Data processing and uncertainty propagation workflow

### 2.3.1. Construction of isotopic maps and blank correction

For each map, the first line corresponds to the ICP-MS signal recorded without laser ablation in order to calculate average blank values for the different masses. The matrices of raw isotopic values (in counts per seconds) were then built using a home-made python code including blank correction (subtraction of average blank for each data point). Visualization as maps were processed with the Fiji freeware. All isotopic maps are imported as an image stack and appropriately cropped. To consider the wash-out time, the number of pixels was averaged by 8 along each linear scan in order to reach a value of one pixel per 500 ms and 1000 ms for He and Ar, respectively. The isotopic ratio matrices were then calculated.

### 2.3.2. Normalization of the Pb/Pb ratios and drift correction of the U/Pb isotopic ratios

For the NIST SRM612 glass, Pb and U isotopic ratios were calculated as the robust mean of the ratios over the entire image, using the "Huber" function of the *Statsmodels* Python library. The robust mean is obtained through an iterative process that uses a Huber loss function to strongly reduce the weight of any outlier (i.e., spike) onto the final result, without any manual outlier rejection. The standard error (i.e., standard deviation of the mean) for each ratio of the NIST SRM612 glass was also calculated from the standard deviation which is an output of the function. For pixel Pb/Pb ratios of the unknown and WC1, normalization (correction of mass bias) was based on a standard bracketing approach, using the NIST SRM612 analyzed immediately before and after the unknown. The reference values were taken from Woodhead and Hergt (2001). For U/Pb ratios, correction for drift was based on a similar approach (standard bracketing using SRM NIST612 glass). For WC1 calcite, an age was calculated from the combination of both isotopic maps that bracket the unknown sample. The age, obtained with the robust regression method described in detail below was used to calculate the correction factors for the U/Pb ratios, following the approach of Roberts et al (2017). The correction factor was calculated for a TW regression, with the $^{207}$Pb/$^{206}$Pb anchored to a value of 0.85 as determined from isotope dilution methods (Roberts et al., 2017). The U/Pb isotopic ratios of all pixels of the unknown sample isotopic maps were corrected by the obtained correction factor.

### 2.3.3. Age calculation of the unknown

All regressions were performed using a statistically robust linear regression approach based on a Tukey bi-weight loss function. All the pixel isotopic values were first plotted in the appropriate plot (TW, 86TW and isochron plots). For the 86TW and isochron plots, the amount of common $^{208}$Pb ($^{208}$Pb$_c$) must be known to calculate the $^{208}$Pb$_c$/$^{206}$Pb, $^{206}$Pb/$^{208}$Pb$_c$, and $^{238}$U/$^{208}$Pb$_c$ ratios. For each pixel, the $^{208}$Pb$_c$ was calculated from the amount of radiogenic $^{208}$Pb obtained from measured $^{232}$Th and the age of precipitation given by the TW plot. This heuristic approach is similar to that proposed by Parrish et al. (2018). The regression was then made using the *lmrob* package in R (Maechler et al, 2019). In the chosen setting (*KS2014*) the package allows to automatically perform an MM-robust regression, which has a high breakdown efficiency and is demonstrated to be highly efficient against outliers (Koller and Stahel, 2011, 2017). Based on an iterative process, the method automatically attributes a

weight to each point, with lowest weights corresponding to the outliers. The final regression is then a weighted linear fit through the points. Finally, the age calculation used the classical geochronological equations, corresponding to the lower intercept of the regression line with the concordia in the TW space, with the x-axis in the TW86 space, and to the slope of the regression line in the isochron diagram.

### 2.3.4. Uncertainties

In the commonly used York-type weighted linear regressions, the weight of each point is inversely proportional to its uncertainty (i.e., size of the ellipse), where the latter commonly corresponds to the quadratic addition of analytical uncertainties (Horstwood et al., 2016). In contrast, the pixel isotopic ratios used for the robust regression have no uncertainty, as they are directly calculated from the number of counts of the relevant masses. All uncertainties (analytical or systematic) were therefore added quadratically to the age uncertainty obtained from the confidence interval of the robust regression. Following recommendations of Horstwood et al. (2016), these uncertainties were added to the U/Pb ratio uncertainty corresponding to the age, rather than to the age uncertainty itself. The analytical uncertainties considered are those calculated for the homogeneous primary reference material (NIST SRM612 glass). Due to the standard bracketing approach used (see above), no excess variance could be calculated. The systematic uncertainties are the decay constant uncertainty of $^{238}U$ (0.1%, 2s), and the $^{238}U/^{206}Pb$ ratio uncertainty of WC1, as estimated by Roberts et al. (2017) (2.7%, 2s). and a long-term excess variance taken as 2% (2s). Due to the low number of sessions carried out with this approach, the long-term variance could not be reliably calculated. This value was therefore chosen by analogy with other studies (e.g., Horstwood et al., 2016; Beaudoin et al., 2018; Guillong et al., 2020), but is likely to change in the future.

### 2.3.5. Estimation of the goodness-of-fit

In geochronology, the parameter usually most used in the estimation of goodness of fit is the Mean Square of Weight Deviates (MSWD), or reduced chi-squared, whose value close to 1 is interpreted as a strong indication of the quality of the regression. The calculation of the MSWD can, by definition, only be performed from values associated with uncertainties, which is not the case for the points used for the robust regression. Two steps have been adopted in this work to quickly assess the goodness of fit of the robust regressions from statistical parameters. The first one the calculation of the F-test of overall significance. If the F-test is higher than the significance level (chosen at 0.05), the regression is rejected and the dating considered unsuccessful. In case of success of this first step, the second one considers several parameters that help assessing the quality of the regression. One parameter is the Residual Standard Error (RSE) of the robust regression, an output of the *lmrob* method, which provides an estimate of the mean dispersion of the points around the regression in absolute value of the y-axis. The smaller the RSE, the smaller the dispersion. Based on our experiments with analyzed carbonate samples, successful samples have an RSE lower than ~12% of the y-intercept (Pb₀) value (i.e., < 0.1 for a common Pb value of 0.85 in TW plots). The second step is the calculation of a MSWD from a discretization of the points used for regression, which we call MSWD-d. For each type of plot, the points are discretized into sets along the regression line obtained with the robust regression. For each set,

the number of points always corresponds to the equivalent of 30 seconds of analysis (i.e. 60 pixels for He and 30 pixels for Ar), which is the most common duration of laser ablations in spot LA-ICP-MS studies. The weighted mean and standard error of the sets are calculated (i.e. ellipses). A MSWD-d value close to 1 is expected for sets well aligned along the regression. Based on our experiments, the ages obtained with MSWD-d values above ~2.5 should be questioned. This approach is close to the one used by Drost et al. (2018) for their age calculations, except that (i) discretization is not performed using the $^{235}U/^{207}Pb$ or $^{238}U/^{208}Pb$ ratios, (ii) the MSWD is not based on the use of York-type regression, and (iii) the number of pixels per ellipse is not adjusted opportunely for each sample to tend toward better MSWD value. In parallel with the calculation of the MSWD-d, a TW plot with the corresponding ellipses, as well as a representation of the running mean with 60 and 30 pixels for He and Ar, respectively, is also systematically produced to assist in the user's assessment of the quality of the point alignment and thus the validity of the linear regressionyuè.

Two additional checks are performed to ensure the validity of the results. First, the ages obtained according to the 3 types of diagrams (TW, 86TW and isochron) must be identical within uncertainty. Second, an age is calculated from the subdivision of the isotope maps into squares, that we call 'pseudo-spots', and that mimic conventional LA-ICP-MS spot analyses. In detail, they have dimensions of 7 x 6 pixels, equivalent to 175 μm x 150 μm and 21 seconds of analysis, except for sample BH14 (October 2018) where they have a dimension of 4 x 6 pixels (100 μm x 150 μm; corresponding to 24 s). The mean and standard error of the $^{238}U/^{206}Pb$ and $^{207}Pb/^{206}Pb$ ratios are calculated from the pixels of each pseudo-spot, and the age is obtained in the TW space using IsoplotR. The age calculated with the robust regression must be, within the uncertainties, identical to that obtained from the pseudo-spots.

### 2.3.6. Colocalization study

The lack of trace element concentration data prevents the adoption of a pixel selection approach as presented by Drost et al. (2018). In order to select the most favourable areas of the isotope maps to obtain precise ages, a pixel value colocalization approach using the ImageJ plugin ScatterJn (Zeitvogel and Obst, 2016) was used. The pixel values of the $^{207}Pb/^{206}Pb$ and $^{238}U/^{206}Pb$ isotopic maps were plotted in a TW scatterplot, and areas selected in the spatial domain (maps) were highlighted in the plot (and vice versa). For all samples analysed, the spatial location of (i) outliers highlighted by the robust regression approach and (ii) visually aligned high $^{238}U/^{206}Pb$ ratio points in the TW plot were identified. In either case (or both), if the identified pixels are located in close spatial proximity on the map, a subset of the map can be selected and the age calculated from the pixels in that subset. It should be noted, however, that due to the small number of pixels selected, the age obtained is not necessarily more precise than using the entire map (as also highlighted by Drost et al. (2018) regarding their approach).

### 2.3.7. Comparison with the method of Drost et al. (2018)

In order to compare the efficiency of the robust regression method with that of Drost et al (2018), we also carried out a processing of isotope ratio data directly inspired by these authors. It consists in sorting the pixel ratio values using the $^{207}Pb/^{235}U$ ratio for pre-Cenozoic samples or the $^{238}U/^{208}Pb$ ratio for Cenozoic samples, clustering the data in discrete steps of a given

number of pixels (here 60 pixels, or 30 pixels for sample BH14, corresponding to 30 s of signal), before calculating the mean and its uncertainty for each cluster. The age is then calculated in TW, 86TW and isochron diagrams using IsoplotR. The stated age uncertainties include the same systematic uncertainties as for the robust regression.

## 3 Results

For all samples considered in this study, the three types of diagrams used for age calculations by the robust regression (TW,
86TW and isochron), the TW plots resulting from the use of the pixel pooling approach Drost et al. (2018) method, and the ones produced from image discretization (pseudo spots), are presented in the Supplementary material (Fig. S3). The ages calculated by the different approaches are also listed in Table 1. Finally, the pixel values of the isotopic images for the masses 238, 232, 208, 207, 206 are presented in Table S1.

### 3.1. Age determination on previously dated samples

**3.1.1. Duff Brown**

The ages were obtained on 2 images chosen as an example. The ones calculated from 'pseudo-spots' in the TW space are imprecise, with values of $70.7 \pm 20.2$ Ma and $54.8 \pm 13.3$ Ma (MSWD of 4.4 and 2.8, respectively; Fig. S3; Table 1). This is due to the lack of spread of the data points (i.e., ellipses) along a linear array, which results in poorly defined slope and intercepts of the regression line. Indeed, anchoring the $Pb_0$ to the value calculated from the data of Hill et al. (2016) ($0.738 \pm$
$0.02$, 2s), leads to more precise ages of $63.6 \pm 2.5$ Ma (MSWD = 4.2) and $66.3 \pm 2.9$ Ma (MSWD = 4.0), similar to the reference value within uncertainties (Fig. S3). The same observations can be proposed for the robust regression method. The ages found without anchoring the common lead value are $54.9 \pm 2.5$ Ma (Fig. 2B, 2D) and $57.4 \pm 2.4$ Ma (Fig. S3) (MSWD-d of 8.5 and 4.1, respectively), whereas using the fixed $Pb_0$ value leads to ages of $63.1 \pm 2.1$ Ma (Fig. 2C) and $65.9 \pm 2.2$ Ma (Fig. S3), similar to the reference value within uncertainties. Note that the maps used for age calculation (Fig. 2A) are smaller ($0.21$ mm$^2$,
$\sim$630 pixels) than those made for other samples, as Duff Brown was used as a validation RM during the tests. The very high MSWD-d values (39.1 and 29.5) are in this case due to the low $^{207}Pb/^{206}Pb$ values of the pixels the closest to the y-axis (i.e., with lower $^{238}U/^{206}Pb$ value) compared to those expected from the anchored regression. Finally, the ages obtained with the method of Drost et al. (2018) are identical to those from the robust regression, within the limits of uncertainties. The ones obtained without anchoring the $Pb_0$ value are $61.9 \pm 3.4$ Ma (MSWD = 3.1; Fig. 2E) and $61.2 \pm 2.5$ Ma (MSWD = 1.3; Fig.
S3), close to the expected value. By anchoring the $Pb_0$ value, the ages are of $63.9 \pm 2.2$ Ma (MSWD = 4.5) and $65.8 \pm 1.0$ Ma (MSWD = 9.1) (Fig. S3).

### 3.1.2. BH14

Two images taken at two distinct locations in the same vein of this sample show a large amount of U (mean = 4 to 11 ppm

depending on the image) (Fig. 3A, 3B), and a very good spread of the pixel U/Pb and Pb/Pb ratios along a linear trend (~2 to 96 for $^{238}U/^{206}Pb$; Fig. 3C, 3D). Such variation in isotope ratios (and U concentrations) is clearly related to crystal growth zoning (Fig. 3A, 3B). This composition allows precise dating, as shown by Beaudoin et al. (2018). Indeed, splitting the isotopic maps in pseudo-spots allows calculating ages of 61.3 ± 3.0 Ma (Fig. 3E) and 61.7 ± 3.0 Ma (Fig. S3), with MSWD of 0.1 and 0.4, respectively, values that are fully consistent with the ages obtained with the robust regressions from the entire maps (Table

1). The latter vary between 60.7 ± 2.0 Ma and 61.8 ± 2.2 Ma according to the diagrams considered (Fig. 3C, 3D; only TW concordia plots shown). They are identical within the limits of uncertainty to that obtained with LA-ICP-MS spot analysis, both by Beaudoin et al. (2018) and in our laboratory (63.2 ± 2.2 Ma and 61.2 ± 2.9 Ma, respectively; Fig. 3F). They are also similar to the ages calculated by using the approach of Drost et al. (2018) (60.8 ± 2.1 Ma to 61.4 ± 2.1 Ma according to the diagrams considered; Fig. 3G and S3). The age uncertainties calculated by the robust regression are generally less than 1 Ma (without considering systematic uncertainties), similar to that obtained by spot analysis and with the method of Drost et al.

(2018). This high precision can be related to the very good visual and statistical parameters calculated for this regression (good point alignment, RSE < 7% of the $Pb_0$ value, and MSWD-d < 1.7 (Fig. 3C, 3D, 3E, 3F)). The colocalization study shows that the points furthest from the trend line of the robust regression are not located in a same area on the isotopic map. Any selection of a part of the map to possibly improve the age calculations is not justified.

### 3.1.3. PXG20-1

This sample is characterized by a moderately high mean U content of 2.5 ppm (Fig 4A), and by a large spread of the pixel U/Pb and Pb/Pb ratios along a linear trend (~0.5 to 93 for $^{238}U/^{206}Pb$) (Fig 4B). This favourable composition explains that the ages obtained by the robust regression are comparable within the uncertainties between the TW concordia plot (40.5 ± 1.6 Ma; Fig. 4B), the 86TW plot (38.2 ± 1.6 Ma; Fig. S3), and the isochron plot (40.2 ± 1.6 Ma; Fig. S3) (Table 1). They are also consistent with the age obtained by LA-ICP-MS spot analysis (40.5 ± 2.3 Ma), both in terms of absolute value and of precision

(Fig. 4D), and with the age calculated from the pseudo-spots (40.5 ± 3.3 Ma; Fig. 4C). The slight variations in the central age depending on the diagram used (38.2 to 40.5 Ma) can be related to the statistical parameters of the regression, which are not as good as those obtained for BH14 (MSWD-d values of 1.1 to 3, and RSE values comprised between 7 and 9% of the $Pb_0$ value). This possibly shows some heterogeneity of isotopic composition, such as for example inclusions of detrital material. The colocalization study shows a globally homogeneous spatial distribution of the values most distant from the regression line

in a TW concordia plot. Any inclusions or cement alteration would certainly be better highlighted using a thresholding approach based on trace element contents, as proposed by Drost et al. (2018). The ages calculated with the method of Drost et al. (2018) for the entire map are within uncertainty of the previous ones: 42.5 ± 1.8 Ma for the TW concordia plot (Fig. 4E), 40.6 ± 1.7 Ma for the 86TW plot (not shown), and 40.7 ± 1.1 Ma for the isochron plot (not shown). Finally, the ages obtained

with both the isotope mapping approaches and LA-ICP-MS spot analysis are centered around ~40 Ma. Such ages are similar
to that suggested by the paleomagnetic study of Oms et al. (2003), which point to a Bartonian age (~40-38.5 Ma). However,
they are slightly older (including within the limits of uncertainty) than the age proposed by Canudo and Molina (1988) on the
basis of the presence of *Globigerinatheka semi-involuta* in the sediments containing the sample. This microfauna rather
indicates an age between 38.5 and 36 Ma according to Premoli-Silva et al. (2006) and Wade et al. (2011), and considering the
GTS2012 (Gradstein et al., 2012). The origin of this discrepancy is not yet identified. Nevertheless, the ages obtained are
consistent, along with the other proxies, with the transition between marine and continental deposits in the studied area being
Bartonian-Priabonian in age.

### 3.1.4. PXG32-2

This lake carbonate sample has a very high U content (mean = 17.5 ppm) (Fig. 5A). However, the distribution of the pixel
U/Pb and Pb/Pb ratios is more restricted than for the PXG-20-1 sample (Fig. 5B). Accordingly, the age calculated from the
pseudo-spots is less precise than for the previous sample (32.5 ± 3.7 Ma) (Fig. 5C). Regarding the robust regression, the ages
obtained are on the other hand all comparable and of greater precision (~35 ± 2.0 Ma, MSWD-d < 1.3 and RSE values lower
than 5% of the $Pb_0$ value) (Fig. 5B and S3; Table 1). The ages obtained with the method of Drost et al. (2018) are slightly
variable, although being similar within the limits of the uncertainties (34.2 ± 1 Ma to 36.6 ± 1.6, MSWD ~1; only the TW
concordia plot is shown on Fig. 5E). Nevertheless, all the previous ages are in agreement with the one obtained by LA-ICP-
MS spot analysis (37 ± 3 Ma; Fig. 5D), and by the paleomagnetic study of Mochales et al. (2016). The age obtained with the
robust regression approach is of higher precision than the age found with LA-ICP-MS spot analysis. However, the age
calculated with spot analysis is associated with unsatisfactory statistics (MSWD = 6.3) and could therefore be considered as
an errorchron. Such high MSWD is clearly linked with sample heterogeneity as seen from the scatter of error ellipses around
the regression line (Fig. 5D). Compositional heterogeneities are also clearly visible with U and Pb elemental maps (Fig. 5A).
This is consistent with the heterogenous nature of the sample, composed of pellets and microbial oncoids (Fig. S2). The
colocalization study indicates that there are no obvious links between such heterogeneities and the location of outlier pixel
values, which are evenly distributed on the maps and do not justify any sub-selection of the isotopic map. The age obtained
confirms the Bartonian age of the first continental deposits in the eastern part of the Jaca Basin in the southern Pyrenees,
bringing an absolute constraint on the end of non-deposition period above the Boltana anticline.

### 3.1.5. JS4 (run 1)

This Carboniferous-Permian lacustrine limestone sample, characterized by high U concentrations (mean ~10 ppm), was
previously dated by Drost et al. (2018). These authors obtained precise ages of at 300.8 ± 3.7 Ma, 299.6 ± 4.0 Ma, and 298.9
± 5.6 Ma for the TW, 86TW and isochron diagrams, respectively. The ages calculated with the robust regression from pixel
U/Pb and Pb/Pb ratios (K. Drost, personal communication) are similar to the previous ones (298.9 ± 3.4 Ma, 297.0 ± 3.4 Ma,

and 300.8 ± 2.8 Ma for the TW, 86TW and isochron diagrams, respectively; only the TW concordia plot is shown on Fig. 6A) (Table 1). The very good statistics of the regressions (MSWD-d ≤ 1, RSE < 5.5% of the $Pb_0$ value) is explained by the large spread of the ratios, defining a clear linear trend in the different plots. Note that to concur with the results given by Drost et al. (2018), these ages are without systematic uncertainties. Adding them leads to final age uncertainties above 8 Ma (Fig. 6A).

### 3.1.6. BM18

Unlike JS4, the BM18 sample, a tectonic calcite vein, is characterized by low U and Pb concentrations (mean of ~200 pb and ~7 ppb, respectively). It was dated precisely to 59.5 ± 2.7 Ma (MSWD = 2.5) by LA-ICP-MS spot analysis, owing to a large, linear spread of the individual ellipses in the TW space (Beaudoin et al., 2018). A similar age (61.0 ± 1.7 Ma in the TW space, MSWD = 1.1) was obtained by Roberts et al. (2019) with the pixel pooling approach of Drost et al. (2018), despite a large scatter of the pixel U/Pb and Pb/Pb isotopic ratios (Fig. 6B). Applying robust regression to these data does not allow the

expected age to be reproduced. When systematic uncertainties similar to those of Beaudoin et al (2018) are considered, the ages obtained are 46.9 ± 2.8 Ma and 48.4 ± 2.5 Ma for the TW (Fig. 6B) and 86TW diagrams (not shown), respectively, more than 10 Ma younger than the expected one (Table 1). In contrast, for the isochron diagram, the age of 60.9 ± 2.1 Ma (not shown) is in line with expectations. Added to high RSE values (> 10% of the $Pb_0$ value), and MSWD-d values ranging from 2.5 to 4.9, these variable ages would lead us to consider the ages as unreliable. The discrepancy between the ages is due to the

presence of pixels with high U/Pb ratios, acting as leverage points for the robust regression in the TW and 86TW diagrams. In order to get closer to the expected age, we added weights to the pixel values used in the robust regression, as is possible with the "lmrob" library. The weights are based on the density of pixels in each plot, as estimated by a Gaussian Kernel Density Estimate (KDE), the bandwidth of which is estimated by Scott's rule (Scott, 1992). In the case of a large dispersion of the data (in particular linked to the counting statistics, even for a sample of homogeneous age), it is expected that the majority of the

points will be grouped along the isochron (i.e. higher density). The use of this approach for the TW and 86TW diagrams gives values similar in uncertainties to that of Beaudoin et al. (2018), but still centered on younger ages (54.4 ± 2.6 Ma and 55.0 ± 2.5 Ma for the TW and 86TW diagrams, respectively; Fig. 6C). However, the ages obtained with the 3 diagrams are not similar within uncertainties, which would again lead to reject the analysis or, as the ages are close to each other, to consider that the real age is probably between ~52 Ma and ~63 Ma. Using the approach of Drost et al. (2018), an age of 60.0 ± 2.8 Ma (MSWD

= 0.95) is obtained for the TW diagram (Fig. S3), i.e. slightly younger than the value presented in Roberts et al. (2019). Doing the same calculation for the TW86 and isochron diagrams (not presented in Roberts et al., 2019) gives values of 55.6 ± 2.9 Ma (MSWD = 1) and 60.1 ± 6.4 Ma (MSWD = 0.5), respectively (not shown). The 3 ages are centered between 55 Ma and 60 Ma, therefore identical to those obtained with the robust regression, but they are moreover similar to each other within uncertainties. Therefore, for this sample of low U and Pb concentrations, the ages calculated with the approach of Drost et al. (2018) are

more reliable than with the robust regression (weighted or not), despite a $Pb_0$ value (~0.7) higher than that expected on the basis of spot analyses (~0.59; Beaudoin et al., 2018).

### 3.2. Unknown samples

### 3.2.1. ETC2

The isotopic map of this sample (Fig. 7A), taken from a tectonic breccia in the Basque Country (France), was made through
several calcite cements as clearly shown in the cathodoluminescence images (Fig. 7B). The latter are characterized by variable
U contents (below the detection limit to > 200 ppm; Fig. 7A). Most pixel have a low U/Pb ratio (< 10), and some dispersion
in the $^{207}Pb/^{206}Pb$ values for the weakly radiogenic pixels can be noticed (Fig. 7C). However, the U/Pb and Pb/Pb ratios of
several pixels define a clear linear trend (Fig. 7C), allowing age calculation. The age obtained from the pseudo-spots is 100.2
± 10.1 Ma (Fig. 7D, Table 1). Those obtained with the robust regression are identical, within the limits of uncertainty: 96.1 ±
3.9 Ma and 96.5 ± 4.0 for the TW (Fig. 7C) and 86TW plots (Fig. S3), respectively, and an age of 95.6 ± 3.3 Ma for the
isochron plot (Fig. S3). These similar ages are consistent with the good statistical parameters obtained from the robust
regression (MSWD-d < 1.2, RSE lower than 8.5% of the $Pb_0$ value). However, both the maps of pixel concentration values
and the colocalization analysis of image data demonstrate that the overdispersion of $^{207}Pb/^{206}Pb$ values are located in a restricted
area of the map corresponding to one cement (cement C1 on the CL image, Fig. 7B), whereas the highest $^{238}U/^{206}Pb$ (i.e., most
385 radiogenic values) that drive the age are located in another restricted area corresponding to another cement (cement C2 on the
CL image) (Fig. 7E). Lead by the colocalization study, the selection of a small area of the map corresponding to cement C2
results in similar but more accurate age calculations than considering the entire map (97.4 ± 3.5 Ma for the TW plot (Fig. 7F),
96.9 ± 3.5 Ma for the 86TW (Fig. S3) and 95.5 ± 3.3 Ma for the isochron plot (Fig. S3)). The statistical parameters are also
better, despite a number of pixels divided by three (MSWD-d < 0.9, RSE lower than 5% of the $Pb_0$ value). This demonstrates
the relevance of the colocalization approach for age calculation with the robust regression approach, especially when the
analytical conditions do not allow trace elements to be analyzed at the same time as U and Pb. In contrast, regarding the
approach of Drost et al. (2018), the ages obtained from the selected area of the map are less precise than those calculated from
the entire map (96.4 ± 6.5 versus 97.9 ± 4.7 Ma for the TW plot (Fig. 7G and S3), 98.0 ± 6.1 versus 100.0 ± 5.0 Ma for the
TW86 plot (not shown), and 95.0 ± 7.5 versus 97.1 ± 5.6 Ma for the TW plot (not shown)). This is explained by the lower
number of ellipses governing the regression.

Finally, in the case of the isotopic image made on the ETC2 sample, the robust regression approach appears to be the most
precise. The age obtained indicates cement precipitation during the rifting event that affected the area before the Iberian-
Eurasian convergence, that led to the formation of the Pyrenees from the Campanian (Mouthereau et al., 2014).

### 3.2.2. ARB

ARB sample is a single, homogenous calcite cement partially filling a tectonic breccia. It has a mean U content of ~1.9 ppm
(Fig. 8A), and a large spread of the pixel U/Pb and Pb/Pb ratios (~0.2 to ~106 for $^{238}U/^{206}Pb$), defining a linear trend (Fig.
8B). Using the robust regression, the calculated ages for this sample are 106.5 ± 3.8 Ma for the TW concordia plot (Fig. 8B),
107.4 ± 3.8 Ma for the 86TW plot (Fig. S3), and 109.7 ± 3.7 Ma for the isochron if the entire image is considered (Fig. S3)

(Table 1). Again, these values are similar within uncertainties, and similar to the age obtained from the pseudo-spots (108.4 ± 8.1 Ma; Fig. 8D). The slight differences in age between the types of plots (< 3%) may be related to the high RSE (9 to 18% of the $Pb_0$ value), which indicates some dispersion around the regression line, despite MSWD-d values lower than 2. Likewise, the ages obtained by the method of Drost et al. (2018) are slightly variable, although consistent with previous ages within the limits of uncertainties (111.0 ± 4.2 Ma for the TW plot (Fig. 8E), 110.6 ± 3.9 for the 86TW plot (not shown), and 105.3 ± 4.6 Ma for the isochron plot (not shown), with MSWDs of 2.6, 2.5 and 1.1, respectively). For the robust regression, the colocalization study indicates that among the more distant points from the regression line in the concordia TW plot, almost all correspond to pixels located in the left quarter of the isotope map, where the U and Pb concentrations are the lowest (Fig. 8A). The age uncertainties obtained without considering these pixels are slightly better than those obtained with the entire map: 107.0 ± 3.8 Ma for the TW concordia plot (Fig. 8C), 107.8 ± 3.8 Ma for the 86TW plot (Fig. S3) and 109.7 ± 3.7 Ma for the isochron plot (Fig. S3). However, contrary to ETC2, the statistical parameters are not better (MSWD-d between 1.4 and 2.4, and RSE between 9.5 and 18% of the $Pb_0$ value) due to the disappearance of several pixels located close to the robust regression line. In contrast, applying the method of Drost et al. (2018) to the map sub-set results in better age uncertainties and MSWD compared to the entire isotopic map, with values of 110.5 ± 4.0 Ma for the TW concordia plot (MSWD = 2; Fig. S3), 110.4 ± 3.8 Ma for the 86TW plot (MSWD = 1.1; not shown) and 104.7 ± 4.4 Ma for the isochron plot (MSWD = 0.6; not shown). Taken together, ETC2 and ARB samples suggest that the emplacement of tectonic breccia in the western Pyrenees was related to extensive syn-rift fault development, rather than to the subsequent tectonic inversion.

### 3.2.3. C6-265-D5

This sample is a dolomite vein from the northern Pyrenees, presumably formed in a tectonic context identical to ETC2 and ARB samples. Its mean U and Pb contents are low (230 ppb and 60 ppb respectively; Fig. 9A), and a large scatter of the U/Pb and Pb/Pb pixel isotope ratios can be evidenced (Fig. 9B). As a result, the age calculated from the pseudo-spots has large uncertainties (115.2 ± 17.4 Ma; Fig. 9C; Table 1), and the use of robust regression does not make it possible to obtain consistent ages between the 3 types of plots (84.5 ± 4.7 Ma, 82.4 ± 4.4 Ma and 105.3 ± 3.6 Ma for the TW, 86TW and isochron plots, respectively; Fig. 8B and S3). These ages are associated with MSWD-d values close to unity, but very high RSE values (21 to 28% of the $Pb_0$ value). Weighting the regression by KDE for the three plots (according to the method presented for sample BM18) allows to resolve these inconsistencies (97.0 ± 5.7 Ma, 94.5 ± 5.3 Ma and 93.7 ± 4.5 Ma for the TW, 86TW and isochron plots, respectively), and to get better statistical parameters (MSWD-d = 0.8 to 1, and RSE = 10 to 21% of the $Pb_0$ value) (Fig. 9D and Fig. S3). However, the new ages are outside uncertainty of those obtained from the pseudo-spots (i.e. younger), which questions their reliability. On the other hand, the ages calculated by the method of Drost et al (2018) are also comparable to that calculated from the pseudo-spots, with values of 110.3 ± 6.2, 102.7 ± 6.2 and 104.7 ± 7.8 for the TW (Fig. 8E), 86TW and isochron plots, respectively, with MSWD values lower than 1.8. It therefore seems that, in a manner analogous to BM18, the results obtained by robust regression for C6-265-D5 are less reliable than those obtained by the pixel-pooling approach of Drost et al. (2018), except for the isochron plots. Nevertheless, although being obtained on dolomite, and therefore

possibly biased by a matrix effect (Elisha et al., 2020), these ages are consistent with those obtained for samples ETC2 and ARB, which once again suggests a link with the rifting event recorded in the Pyrenees during the Cretaceous.

## 4 Discussion

### 4.1. Interest of dating from isotope maps

The pioneering work of Drost et al. (2018) has shown the obvious value of the isotope map dating approach, where data on minor and trace element concentrations are used to select the most suitable pixels for age determination. This initial step seems really essential and should be followed up systematically if the analytical conditions makes it possible (e.g. use of an ICP-MS with a fast mass scan such as a quadrupole or TOF). If not, the operator must remain attentive to the quality of the data, by checking the scattering of the pixels around the regression lines and the possible presence of phase mixtures. The use of a colocalization approach as presented in our study can be an alternative way, although less efficient, to carry out this careful study of the spatial distribution of the furthest pixel values from the regression line. The examples presented above, in particular that of the ETC2 sample, however show that colocalization is more suited to the robust regression approach than to that of pixel pooling. It is in any case fundamental to carry out an initial petrographic characterization of the samples as recalled by Roberts et al. (2019). Taking these precautions into account, our results and those of Drost et al. (2018) and Roberts et al. (2019) show that the ages obtained from the image mapping approach are perfectly comparable, both in value and uncertainty, with those obtained from more common methods (LA-ICP-MS spot analysis and isotope dilution). Additionally, it should be pointed out that this approach has several advantages in comparison with the more common ones. First, the image mapping approach is particularly suitable for easily dating multiple generations of cements filling porosities or fractures. Such an approach is fast and simple to set up, since it is sufficient to map a large portion of the sample under consideration to cover several generations of cements with number of pixels high enough for age calculation, and then to select the pixels (preferentially by a geochemical approach) corresponding to the different cements to calculate the ages. This methodology has been carried out successfully on sample ARB with colocalization. Second, contrary to spot analyses, the simultaneous mapping of trace elements useful both for geochemical characterization and for dating makes it possible to gather all the necessary information in a single analysis, without changing the analytical parameters of the LA-ICP-MS, which may save a considerable amount of time. Third, a complete isotope map can be acquired very quickly (less than 30 minutes, without the reference materials). This allows to consider the analysis of many samples over a limited number of analytical sessions. It is therefore likely that this method to U-Pb dating of carbonates will develop rapidly in the geochronology community, alongside with the now well-established LA-ICP-MS spot analysis approach.

### 4.2. Benefits and limitations of the robust regression to U-Pb image dating

The data processing method presented here, namely the robust regression through the isotope ratio values of the pixels represented in TW or isochron diagrams, is simple and allows a fast age determination and its uncertainty. We recall, as already

mentioned previously, the importance of a preliminary selection of these pixels on the basis of the concentrations of trace elements of the carbonates analysed. As an alternative to the data pooling approach presented by Drost et al. (2018), it has the advantage of not involving the operator in the choice of the number of pixels in each 'pseudoanalysis', since all pixels are considered for a single regression. Their representation in the appropriate diagrams gives a quick idea of the quality of the data, assisted by statistical parameters such as RSE or MSWD-d. Note that the interest of using robust regression for U-Pb dating was already underlined by Powell et al. (2002) and very recently by Powell et al. (2020). It is also implemented in the famous Isoplot software (Ludwig, 2012). Ludwig (2012) stressed the precautions to be taken when using robust regression to construct isochrons, since the absence of uncertainties on each point makes it impossible to distinguish the share of scatter around the regression due to "geological/geochemical complications" from that due to analytical errors alone. In conventional spot analyses, the MSWD calculation is an efficient way to determine the share of each source of uncertainty. Its use in the context of dating from isotopic images, whether with the approach of Drost et al. (2018) or our own, does not solve this problem since the MSWD is calculated from an "artificial" grouping of pixels whose associated ellipses do not represent analytical uncertainties. At most, the MSWD (or MSWD-d) is here a means of verifying the correct alignment of values in the TW and isochron diagrams, with values close to 1 indicating good alignment, information that can be complemented by the moving average calculation as we propose, among other possibilities. For example, the very recent approach proposed by Powell et al. (2020), with calculation of a normalized (robust) median of the absolute values of the residuals, could provide a suitable means to distinguish isochrons from errorchrons. It still has to be implemented in our data reduction scheme.

Among the negative points of the robust regression, the example of samples BM18 and C6-265-D5 shows the sensitivity of the method to the dispersion of the U/Pb and Pb/Pb ratios. For this type of low-U sample, where noisy data are linked to low counting statistics, the ages calculated without weighting are too young for the TW and 86TW plots. They are also inconsistent between the TW, 86TW and isochron plots. This would lead to rejecting samples which, with the method proposed by Drost et al. (2018), are validated by obtaining consistent ages not only between the different diagrams, but also with those obtained by spot or by 'pseudo-spots' analyses. Weighting the robust regression by KDE improves the results by giving consistent ages between the diagrams. However, the comparison with the expected ages shows that the ages calculated in this way are too young by around 5 to 10% (55-60 Ma instead of 60 Ma for BM18, and 94-97 Ma instead of 118 Ma for C6-265-D5). For such kind of samples, the tests carried out show the greatest reliability of the method of Drost et al. (2018) for image-based dating. However, as this latter approach might give erroneous $Pb_0$ values (example of BM18), we believe that LA-ICP-MS spot analyses should be preferred when U concentrations are far below ppm.

**5 Conclusion**

U-Pb LA-ICP-MS carbonate dating from isotope images is an extremely recent approach which, compared to more common dating methods such as LA-ICP-MS spot analyses, has the great advantage of allowing the use of images for both sample screening and dating. In this contribution, U-Pb ages of several carbonate samples were obtained from micro-scale LA-ICP-

MS isotopic maps, using a new, alternative data processing method based on robust regression. The method, which is very simple to set up, consists of extracting the U/Pb and Pb/Pb pixel values from the image, plotting them in a diagram used for dating such as a concordia diagram, and performing robust regression to calculate the age. Prior to dating, we also conducted a colocalization study of the sample isotope ratios that was used to locate the most suitable areas to age determination. For most samples that had already been dated by U-Pb geochronology using conventional LA-ICP-MS spot analysis or isotope dilution, the robust regression approach method provided similar ages and sometimes better uncertainties compared to both others methods, with an ablation duration less than 40 min. For low-U samples with noisy U/Pb and Pb/Pb pixel values, however, the ages were biased towards too young values by 5 to 10%. In this case, the already existing image-based dating approach of Drost et al. (2018) gave better results. The robust regression processing of LA-ICP-MS isotope map data thus appears to be a simple means to proceed to carbonate U-Pb geochronology, provided that a careful examination of the pixel ratio data is performed beforehand.

## 6 Appendix

### 6.1 Analytical conditions

| Laboratory & Sample Preparation | | |
|---|---|---|
| Laboratory name | Institut des sciences analytiques et de physico-chimie pour l'environnement et les matériaux (IPREM), UPPA, Pau (France) | |
| Sample type/mineral | Calcite | |
| Sample preparation | In situ in polished blocks | |
| Imaging | Two per sample (one for sample screening and one for dating) | |
| **Laser ablation system** | | |
| Make, Model & type | Lambda 3, Nexeya (France) | |
| Ablation cell | Home-made (home-designed) two volumes ablation cell. The large cell has a rectangular shape and a volume of 11.25 cm$^3$ (75 x 25 x 6 mm size) while the small one, placed above the sample is of 10 mm diameter. | |
| Laser wavelength (nm) | 257 nm | |
| Pulse duration (fs) | 360 fs | |
| Fluence (J.cm$^{-2}$) | 5-8 J.cm$^{-2}$ | |
| | *Preliminary imaging* | *Isotope dating maps* |
| Repetition rate (Hz) | 300 Hz | 500 Hz |
| Gas blank (s) | No | 42 s to 74 s per image (1 line) |
| Ablation duration (s) | | 19 min to 38 min |
| Washout and/or travel time in between analyses (s) | Wash out time: 500 ms (He) or 1000 ms (Ar). 15 s of break between lines to allow data processing | |
| Spot diameter (μm) | 17 μm | |
| Sampling mode / pattern | Ablation lines (50 μm-width) made by combining laser beam movement across | Ablation lines (25 μm-width) made by combining laser beam movement across the surface (5 mm/s) and stage movement (25 μm/s). 25 μm between lines. |

| | | |
|---|---|---|
| | the surface (10 mm/s) and stage movement (100 μm/s). 100 μm between lines. | |
| Cell Carrier gas (L/min) | Ar = 0.650-0.700 L/min (october 2018) or He = 0.600 L/min (April 2019, October 2019) | |
| **ICP-MS Instrument** | | |
| Make, Model & type | ICP-MS Thermo Fisher Element2 HR Jet Interface | |
| RF power (W) | 1000 - 1100W | |
| Cooling gas flow rate | 16 L min$^{-1}$ | |
| Auxiliary gas flow rate | 1 L min$^{-1}$ | |
| Nebuliser gas flow rate | 0.5 L min$^{-1}$ | |
| Masses measured | 206,207,238 | 206, 207, 208, 232, 238 |
| Samples per peak | 40 | 30 |
| Mass window | 50 % | 10 % |
| Sample time | 8.3 ms | 3 ms |
| Settling time | 1 ms | 1 ms |
| Mass sweep | 500 ms | 60 ms |
| Averaged mass sweep | 2 | 8 |
| Resolution | 300 | |
| Sensitivity | Percentage of ions detected with regard to atoms ablated is ~0.04% for U, as calculated with NIST 612 with the IC-PMS method used for the dating map (comprising 6 isotopes) | |
| **Data Processing** | | |
| Calibration strategy | No | Calibration by standard bracketing; NIST614 (October 2018) and 612 (April, October 2019) for Pb-Pb and WC-1 calcite for Pb-U |
| Reference Material info | No | Primary: NIST614 / 612 - Woodhead and Hergt (2001) WC-1 254.4 ± 6.4 Ma (2s) - Roberts et al., 2017 Validation: Duff Brown 64.04 +/- 0.67 Ma (2s) - Hill et al., 2016 |
| Data processing package used / Correction for LIEF | Element XR acquisition software, data processing with in-house Python/R code and ImageJ software. | |
| Common-Pb correction, composition and uncertainty | No | No common Pb correction. Unanchored robust regressions in Tera-Wasserburg, 86TW plot (Parrish et al., 2018) and $^{206}Pb/^{208}Pb_{common}$ versus $^{238}U/^{208}Pb_{common}$ isochron plots. Ages in the figures are quoted at 95% absolute uncertainties and include systematic uncertainties, propagation is by quadratic addition. |
| Quality control / Validation | No | 5 analyses of Duff Brown (anchored to common Pb value of 0.738) gave ages of 66.87 ± 2.48 Ma (April 2019), 67.79 ± 2.38 Ma, 65.76 ± 2.30 Ma, 64.67 ± 2.29 Ma and 63.06 ± 2.17 Ma (October 2019) |

## 7. Data availability

The methodology for LA-ICP-MS spot analyses, the screening maps, and the plots used for age calculations are presented in the Supplementary material in pdf format. The pixel values of the isotopic images for the masses 238, 232, 208, 207, 206 are presented in the supplementary Table (Table S1, excel format).

## 8. Supplement link

## 9 Author contribution

GH, FC, CPe designed the experiments. GH, FC, CPe, PAG, GM carried them out. GH and CPa developed the model code and performed the simulations. OC and JPG supervised the project and provided the financial support for most experiments. GH prepared the manuscript with contribution from all co-authors.

## 10 Competing interests

The authors declare that they have no conflict of interest.

## 11 Acknowledgements

Gaëlle Barbotin is thanked for help during the analytical sessions. This work benefited from funding from the Université de Pau et Pays de l'Adour (AAP Incitatif Recherche) and from Total (project DAC). The initial manuscript was greatly improved owing to the comments of two anonymous reviewers.

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

| Sample | Method | Plot | Age (without / with systematic uncertainty) | Pb₀ | MSWD or d-MSWD | RSE | RSE (% of Pb₀ value) |
|---|---|---|---|---|---|---|---|
| **Duff Brown 1 (unanchored)** | **Pseudo-spots** | *TW* | 54.76 ± 13.16 / 13.29 | 0.572 | 2.83 | | |
| | **Robust regression** | *TW* | 57.41 ± 1.41 / 2.37 | 0.61 | 4.06 | 0.03122 | 5.12% |
| | | *86TW* | 58.31 ± 1.26 / 2.31 | 1.604 | 5.54 | 0.08204 | 5.11% |
| | | *Isochron* | 62.29 ± 1.26 / 2.42 | 0.572 | 0.46 | 0.06943 | 12.14% |
| | **Drost** | *TW* | 61.20 ± 1.49 / 2.53 | 0.658 | 1.33 | | |
| | | *86TW* | 61.41 ± 2.07 / 2.92 | 1.717 | 2.34 | | |
| | | *Isochron* | 62.17 ± 2.62 / 3.35 | 0.572 | 0.44 | | |
| **Duff Brown 1 (anchored to Pb₀)** | **Pseudo-spots** | *TW* | 66.28 ± 1.83 / 2.87 | 0.738 | 4.02 | | |
| | **Robust regression** | *TW* | 65.94 ± 0.46 / 2.23 | 0.738 | 29.53 | 0.03736 | 5.06% |
| | **Drost** | *TW* | 65.81 ± 1.00 / 2.4 | 0.738 | 9.07 | | |
| **Duff Brown 2 (unanchored)** | **Pseudo-spots** | | 70.75 ± 20.05 / 20.19 | 0.98 | 4.39 | | |
| | **Robust regression** | *TW* | 54.89 ± 1.72 / 2.51 | 0.592 | 8.48 | 0.02746 | 4.64% |
| | | *86TW* | 54.07 ± 1.72 / 2.49 | 1.388 | 8.10 | 0.06897 | 4.97% |
| | | *Isochron* | 62.81 ± 1.46 / 2.55 | 0.569 | 1.03 | 0.07538 | 13.25% |
| | **Drost** | *TW* | 61.91 ± 2.67 / 3.34 | 0.710 | 3.14 | | |
| | | *86TW* | 60.03 ± 2.59 / 3.28 | 1.624 | 2.33 | | |
| | | *Isochron* | 62.64 ± 4.11 / 4.61 | 0.574 | 0.73 | | |
| **Duff Brown 2 (anchored to Pb₀)** | **Pseudo-spots** | *TW* | 63.64 ± 1.24 / 2.46 | 0.738 | 4.22 | | |
| | **Robust regression** | *TW* | 63.05 ± 0.33 / 2.13 | 0.738 | 37.05 | 0.03087 | 4.18% |
| | **Drost** | *TW* | 63.18 ± 0.45 / 2.16 | 0.738 | 3.22 | | |
| **BH14 - Image 1** | **Pseudo-spots** | *TW* | 61.26 ± 2.20 / 3.01 | 0.720 | 0.086 | | |
| | **Robust regression** | *TW* | 61.59 ± 0.34 / 2.07 | 0.733 | 1.67 | 0.02219 | 3.03% |
| | | *86TW* | 61.33 ± 0.33 / 2.06 | 1.837 | 1.54 | 0.05626 | 3.06% |
| | | *Isochron* | 60.67 ± 0.19 / 2.02 | 0.542 | 1.18 | 0.03690 | 6.81% |
| | **Drost** | *TW* | 61.91 ± 0.40 / 2.11 | 0.736 | 1.87 | | |
| | | *86TW* | 61.69 ± 0.40 / 2.11 | 1.846 | 1.94 | | |
| | | *Isochron* | 60.82 ± 0.45 / 2.08 | 0.541 | 0.92 | | |
| **BH14 - Image 2** | **Pseudo-spots** | *TW* | 61.67 ± 2.83 / 3.50 | 0.735 | 0.433 | | |
| | **Robust regression** | *TW* | 61.72 ± 0.68 / 2.17 | 0.737 | 0.97 | 0.02173 | 2.95% |
| | | *86TW* | 61.84 ± 0.72 / 2.19 | 1.853 | 0.76 | 0.06287 | 3.39% |
| | | *Isochron* | 61.45 ± 0.58 / 2.13 | 0.538 | 1.21 | 0.02358 | 4.38% |
| | **Drost** | *TW* | 61.66 ± 0.97 / 2.28 | 0.737 | 2.45 | | |
| | | *86TW* | 62.06 ± 0.84 / 2.24 | 1.854 | 1.89 | | |
| | | *Isochron* | 60.83 ± 0.85 / 2.21 | 0.539 | 1.30 | | |
| **PXG20-1** | **Pseudo-spots** | *TW* | 40.49 ± 3.02 / 3.31 | 0.823 | 0.54 | | |
| | **Robust regression** | *TW* | 40.53 ± 0.85 / 1.59 | 0.823 | 2.54 | 0.054 | 6.56% |
| | | *86TW* | 38.22 ± 0.88 / 1.55 | 2.040 | 3.04 | 0.14728 | 7.22% |
| | | *Isochron* | 40.19 ± 0.85 / 1.58 | 0.485 | 1.06 | 0.04453 | 9.18% |

| | | | | | | | |
|---|---|---|---|---|---|---|---|
| | **Drost** | *TW* | 42.52 ± 1.07 / 1.78 | 0.830 | 2.10 | | |
| | | *86TW* | 40.66 ± 1.13 / 1.77 | 2.066 | 2.46 | | |
| | | *Isochron* | 40.66 ± 1.13 / 1.77 | 0.485 | 1.62 | | |
| | **Pseudo-spots** | *TW* | 32.54 ± 3.57 / 3.73 | 0.797 | 1.35 | | |
| **PXG32-2** | **Robust regression** | *TW* | 34.90 ± 0.80 / 1.41 | 0.809 | 1.53 | 0.03050 | 3.77% |
| | | *86TW* | 34.83 ± 0.82 / 1.42 | 1.929 | 1.20 | 0.07760 | 4.02% |
| | | *Isochron* | 34.70 ± 0.69 / 1.35 | 0.511 | 0.96 | 0.02562 | 5.01% |
| | **Drost** | *TW* | 36.07 ± 1.01 / 1.58 | 0.814 | 1.70 | | |
| | | *86TW* | 36.57 ± 1.05 / 1.61 | 1.950 | 2.20 | | |
| | | *Isochron* | 34.18 ± 0.99 / 1.51 | 0.512 | 1.40 | | |
| **JS4** | **Robust regression** | *TW* | 298.87 ± 3.38 / 8.59 | 0.856 | 1.00 | 0.02954 | 3.45% |
| | | *86TW* | 296.98 ± 3.67 / 8.66 | 2.088 | 0.55 | 0.08372 | 4.01% |
| | | *Isochron* | 300.85 ± 2.80 / 8.42 | 0.478 | 0.62 | 0.02509 | 5.25% |
| **BM18** | **Robust regression (unweighted)** | *TW* | 46.87 ± 2.32 / 2.64 | 0.562 | 4.68 | 0.16692 | 29.70% |
| | | *86TW* | 48.38 ± 1.97 / 2.36 | 1.400 | 4.91 | 0.38233 | 27.31% |
| | | *Isochron* | 60.86 ± 0.63 / 1.75 | 0.612 | 2.48 | 0.3566 | 58.27% |
| | **Robust regression (weighted)** | *TW* | 54.41 ± 1.85 / 2.36 | 0.572 | 1.33 | 0.08479 | 14.82% |
| | | *86TW* | 54.96 ± 1.63 / 2.20 | 1.428 | 2.42 | 0.2138 | 14.97% |
| | | *Isochron* | 64.88 ± 0.96 / 1.99 | 0.576 | 2.49 | 0.2954 | 51.28% |
| **ARB1** | **Pseudo-spots** | *TW* | 108.42 ± 7.16 / 8.02 | 0.838 | 0.28 | | |
| | **Robust regression** | *TW* | 106.51 ± 1.40 / 3.82 | 0.824 | 1.99 | 0.08599 | 10.44% |
| | | *86TW* | 107.42 ± 1.20 / 3.78 | 2.028 | 1.69 | 0.19409 | 9.57% |
| | | *Isochron* | 109.71 ± 0.20 / 3.66 | 0.479 | 1.13 | 0.08979 | 18.75% |
| | **Drost** | *TW* | 111.05 ± 1.87 / 4.15 | 0.845 | 2.59 | | |
| | | *86TW* | 110.61 ± 2.1 / 4.24 | 2.062 | 2.53 | | |
| | | *Isochron* | 105.33 ± 2.94 / 4.58 | 0.488 | 1.09 | | |
| **ARB1 (colocalization)** | **Robust regression** | *TW* | 107.03 ± 1.35 / 3.82 | 0.828 | 2.40 | 0.08049 | 9.72% |
| | | *86TW* | 107.85 ± 1.17 / 3.78 | 2.036 | 2.05 | 0.18238 | 8.96% |
| | | *Isochron* | 109.69 ± 0.20 / 3.66 | 0.478 | 1.41 | 0.08440 | 17.66% |
| | **Drost** | *TW* | 110.52 ± 1.48 / 3.97 | 0.842 | 2.04 | | |
| | | *86TW* | 110.42 ± 1.07 / 3.83 | 2.058 | 1.73 | | |
| | | *Isochron* | 104.72 ± 2.67 / 4.40 | 0.488 | 0.61 | | |
| **ETC2** | **Pseudo-spots** | *TW* | 100.24 ± 9.48 / 10.05 | 0.830 | 1.10 | | |
| | **Robust regression** | *TW* | 96.14 ± 2.21 / 3.89 | 0.828 | 0.87 | 0.05712 | 6.90% |
| | | *86TW* | 96.49 ± 2.33 / 3.97 | 2.038 | 0.90 | 0.15650 | 7.68% |
| | | *Isochron* | 95.64 ± 0.94 / 3.33 | 0.488 | 1.19 | 0.04089 | 8.38% |
| | **Drost** | *TW* | 97.91 ± 3.43 / 4.74 | 0.830 | 0.72 | | |
| | | *86TW* | 99.96 ± 3.71 / 4.99 | 2.045 | 0.35 | | |
| | | *Isochron* | 97.13 ± 4.61 / 5.64 | 0.488 | 0.90 | | |
| **ETC2 (colocalization)** | **Robust regression** | *TW* | 97.44 ± 1.41 / 3.54 | 0.827 | 0.88 | 0.02902 | 3.51% |
| | | *86TW* | 96.93 ± 1.36 / 3.51 | 2.032 | 0.70 | 0.07409 | 3.65% |
| | | *Isochron* | 95.52 ± 0.67 / 3.26 | 0.491 | 0.68 | 0.02356 | 4.80% |
| | **Drost** | *TW* | 96.39 ± 5.64 / 6.49 | 0.828 | 1.31 | | |

| | | | | | | | |
|---|---|---|---|---|---|---|---|
| | | *86TW* | 97.97 ± 5.10 / 6.06 | 2.038 | 0.61 | | |
| | | *Isochron* | 95.0 ± 6.82 / 7.52 | 0.490 | 0.67 | | |
| | **Pseudo-spots** | *TW* | 115.15 ± 17.01 / 17.44 | 0.781 | 0.82 | | |
| | **Robust regression (unweighted)** | *TW* | 84.46 ± 3.75 / 4.68 | 0.715 | 3.87 | 0.15335 | 21.45% |
| | | *86TW* | 82.37 ± 3.41 / 4.37 | 1.739 | 3.82 | 0.36569 | 21.03% |
| | | *Isochron* | 105.35 ± 0.73 / 3.57 | 0.532 | 0.93 | 0.15160 | 28.50% |
| **C6-265-D5** | **Robust regression (weighted)** | *TW* | 97.02 ± 4.76 / 5.74 | 0.740 | 1.02 | 0.07619 | 10.30% |
| | | *86TW* | 94.50 ± 4.34 / 5.35 | 1.792 | 0.95 | 0.18238 | 10.18% |
| | | *Isochron* | 93.75 ± 3.22 / 4.48 | 0.546 | 0.81 | 0.11667 | 21.37% |
| | **Drost** | *TW* | 110.31 ± 5.09 / 6.28 | 0.770 | 1.83 | | |
| | | *86TW* | 102.74 ± 5.18 / 6.21 | 1.838 | 1.67 | | |
| | | *Isochron* | 104.69 ± 7.25 / 8.05 | 0.539 | 1.05 | | |

**Table 1: Ages calculated from the robust regression, from the 'pseudo-spots', and from the approach of Drost et al. (2018), for all samples.**

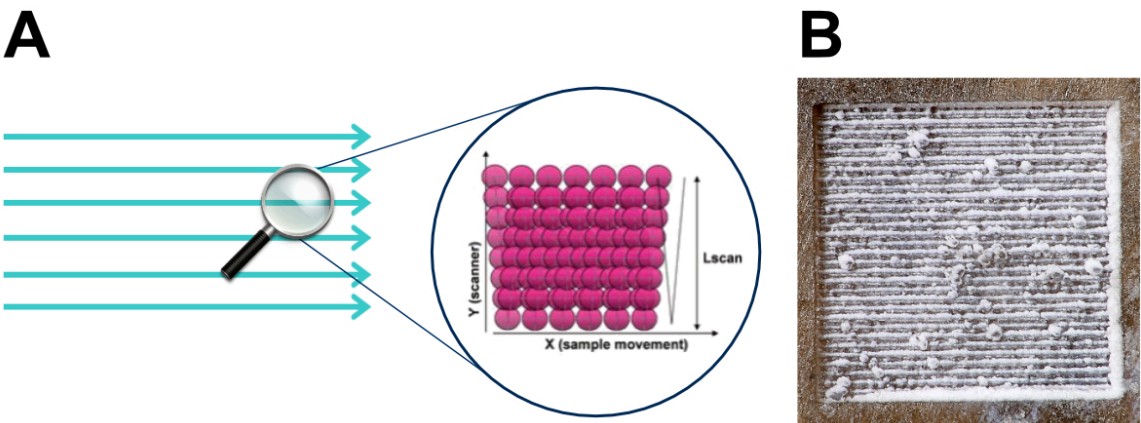

**Figure 1: A: Principle of the construction of the ablation lines with the femtosecond laser used in the study. Lscan corresponds to the width of the line, here 50 μm or 25 μm. B: Example of ablation produced by the rasters used for producing an isotopic map (dimensions: 800 μm x 800 μm).**

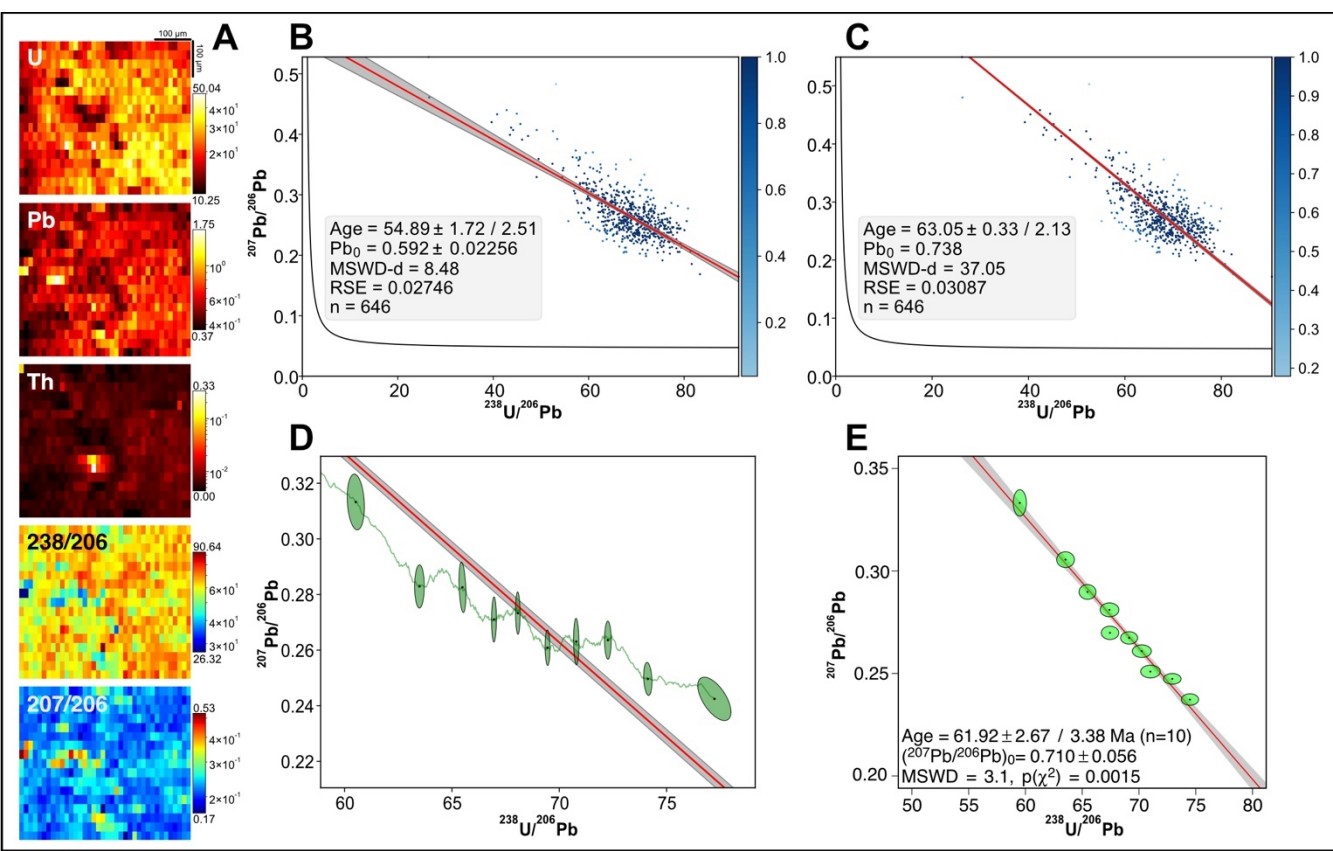

**Figure 2: A:** Element concentrations in ppm (U, Pb, Th) and $^{238}U/^{206}Pb$, $^{207}Pb/^{206}Pb$ isotopic maps obtained for one analysis of Duff Brown sample. Note that the concentrations are normalized to the mean number of counts measured in the NIST reference material for the considered masses, and should thus be regarded only as semi-quantitative. **B:** TW concordia plot with age calculated by robust regression, without anchoring the common Pb ($Pb_0$) value. The age uncertainties ($1.96\sigma$ level) are without (left) and with uncertainty propagation (right). The blue color scale refers to the weight attributed to each pixel by the robust regression. **C:** Same as B but with common Pb anchored to a value of 0.738 as calculated from Hill et al. (2016). **D:** TW concordia plot with ellipses calculated from the sets obtained by the discretization of the pixel ratio values used for regression, and used for calculation of the MSWD-d (see text for details). The regression and associated confidence interval are those obtained by the robust regression. In this example the $Pb_0$ value is anchored. **E:** TW concordia plot of the same sample obtained with the method of Drost et al. (2018), without anchoring the common Pb value. Age and associated uncertainties were obtained from York-type regression using IsoplotR (age uncertainties ($1.96\sigma$ level) are without (left) and with systematic uncertainty (right)).

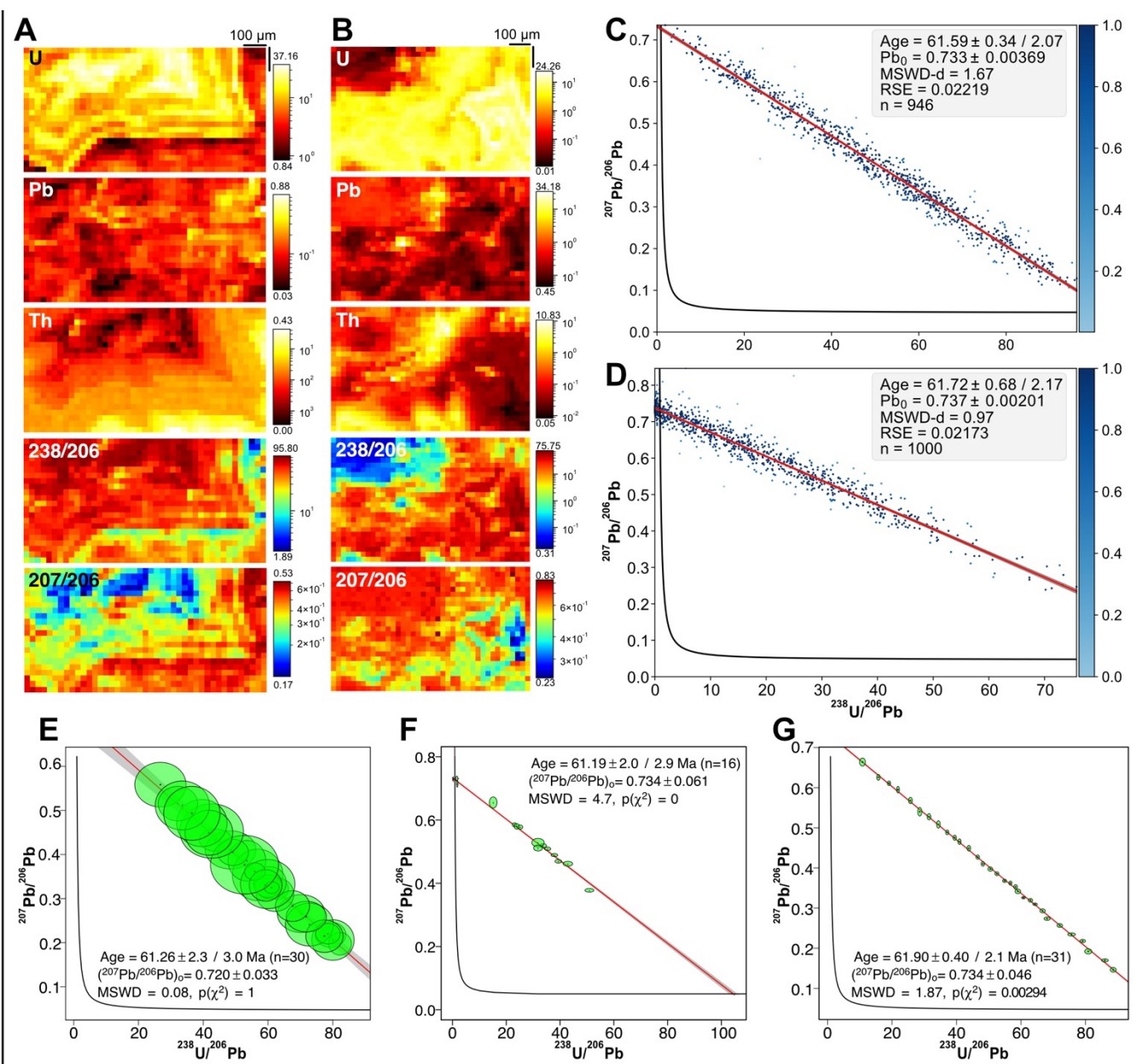

**Figure 3: Results obtained for two isotopic maps of sample BH14 (A, C, E, G: first map; B, D: second map). A, B: Semi-quantitative element concentrations in ppm (U, Pb, Th) and $^{238}U/^{206}Pb$, $^{207}Pb/^{206}Pb$ isotopic maps. C, D: TW concordia plots with ages and their uncertainty calculated by robust regression. E: TW concordia plot with age and its uncertainty calculated from discretization of the isotopic map ('pseudo-spots'). F: TW concordia plot obtained from LA-ICP-MS spot analyses. G: TW concordia plot obtained with the approach of Drost et al. (2018). For E, F, G and similar plots in the next figures, ages and associated uncertainties were obtained from York-type regression using IsoplotR (age uncertainties (1.96σ level) are without (left) and with systematic uncertainty (right)).**

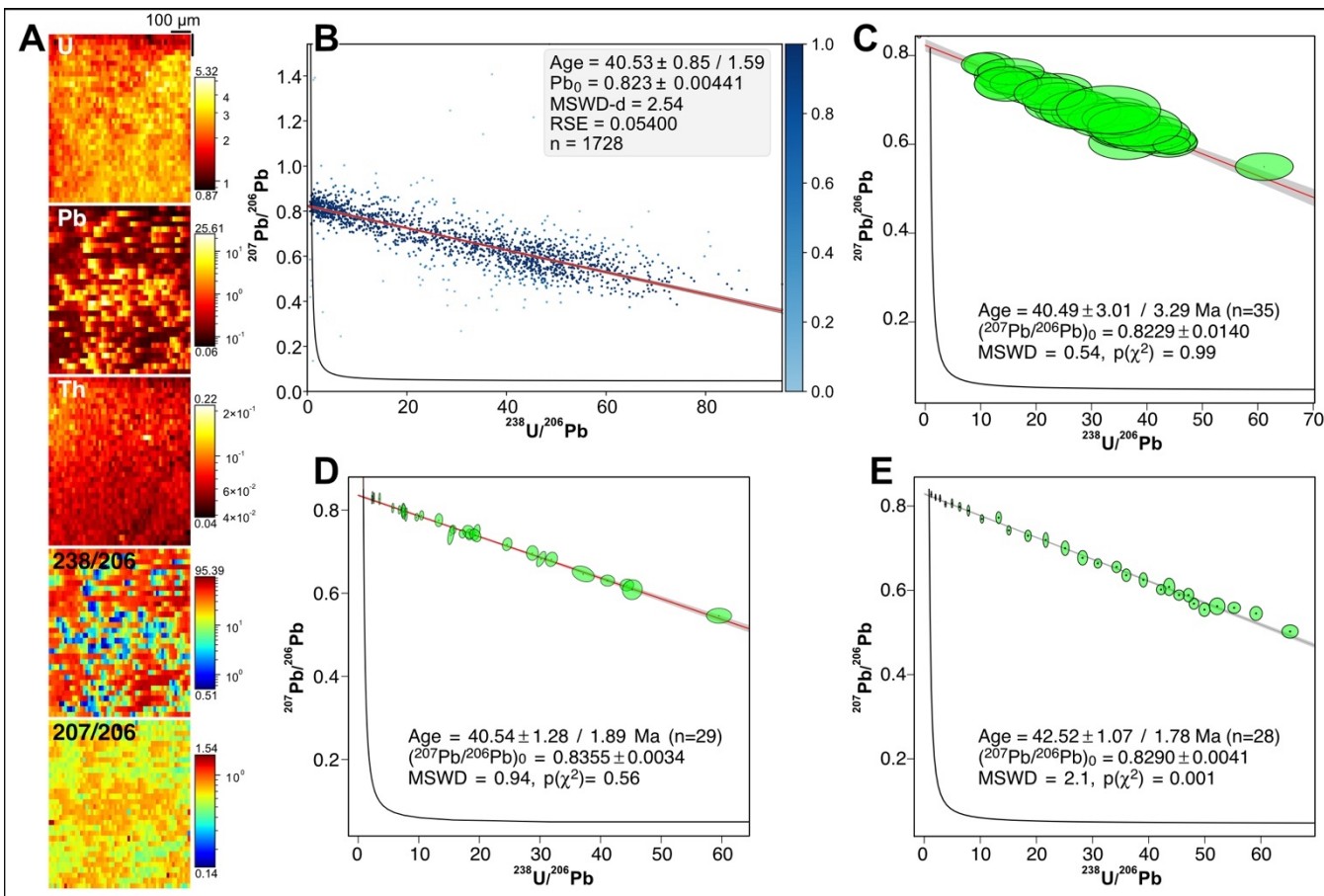

**Figure 4: Results obtained for sample PXG20-1. A: Semi-quantitative element concentrations in ppm (U, Pb, Th) and $^{238}$U/$^{206}$Pb, $^{207}$Pb/$^{206}$Pb isotopic maps. B: TW plot with age and its uncertainty calculated by robust regression. C: TW concordia plot calculated from discretization of the isotopic map ('pseudo-spots'). D: TW concordia plot obtained from LA-ICP-MS spot analyses. E: TW concordia plot obtained with the approach of Drost et al. (2018).**

650

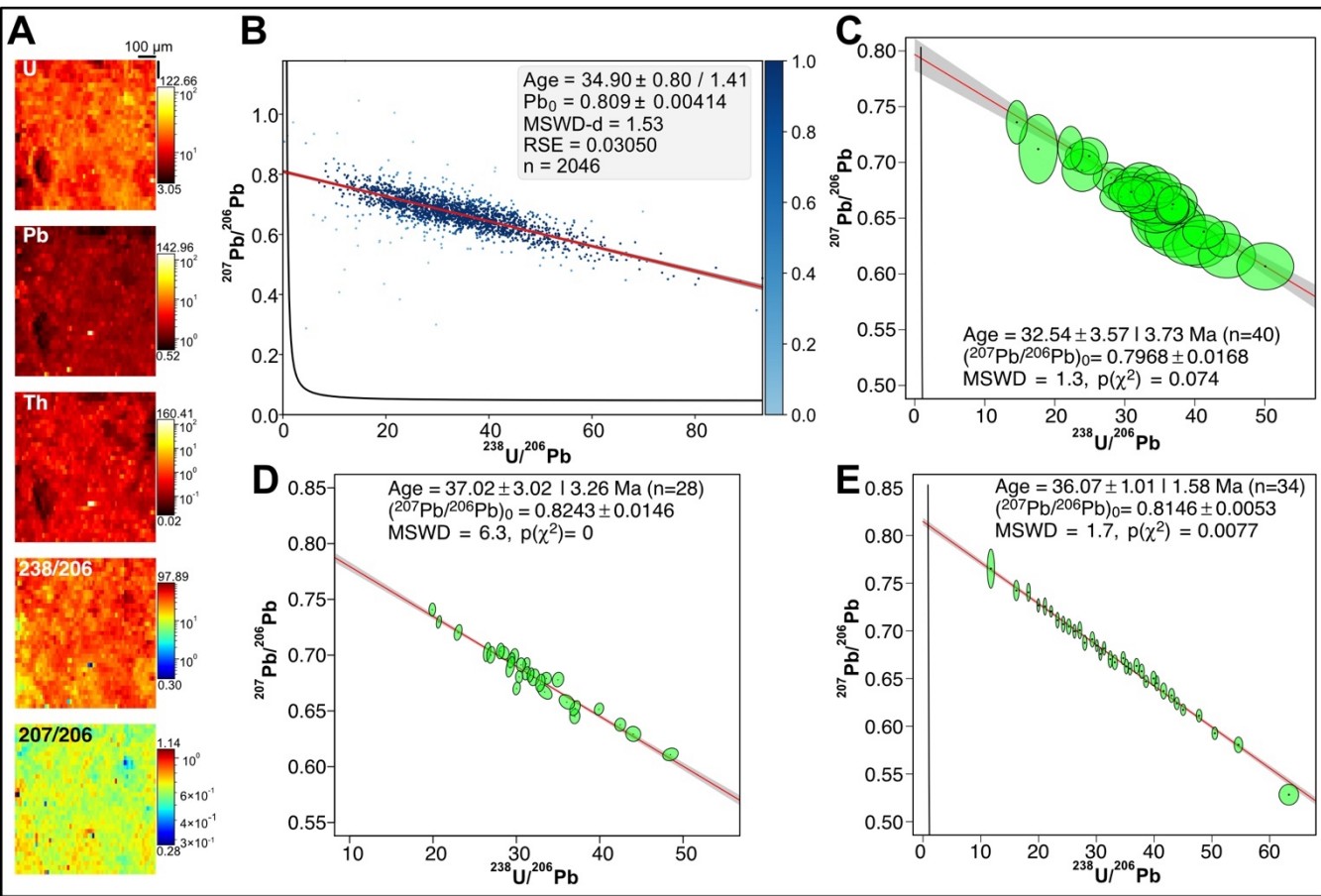

**Figure 5: Results obtained for sample PGX32-2. A: Semi-quantitative element concentrations in ppm (U, Pb, Th) and $^{238}$U/$^{206}$Pb, $^{207}$Pb/$^{206}$Pb isotopic maps. B: TW concordia plot calculated by robust regression. C: TW concordia plot calculated from discretization of the isotopic map ('pseudo-spots'). D: TW concordia plot obtained from LA-ICP-MS spot analyses. E: TW concordia plot obtained with the approach of Drost et al. (2018).**

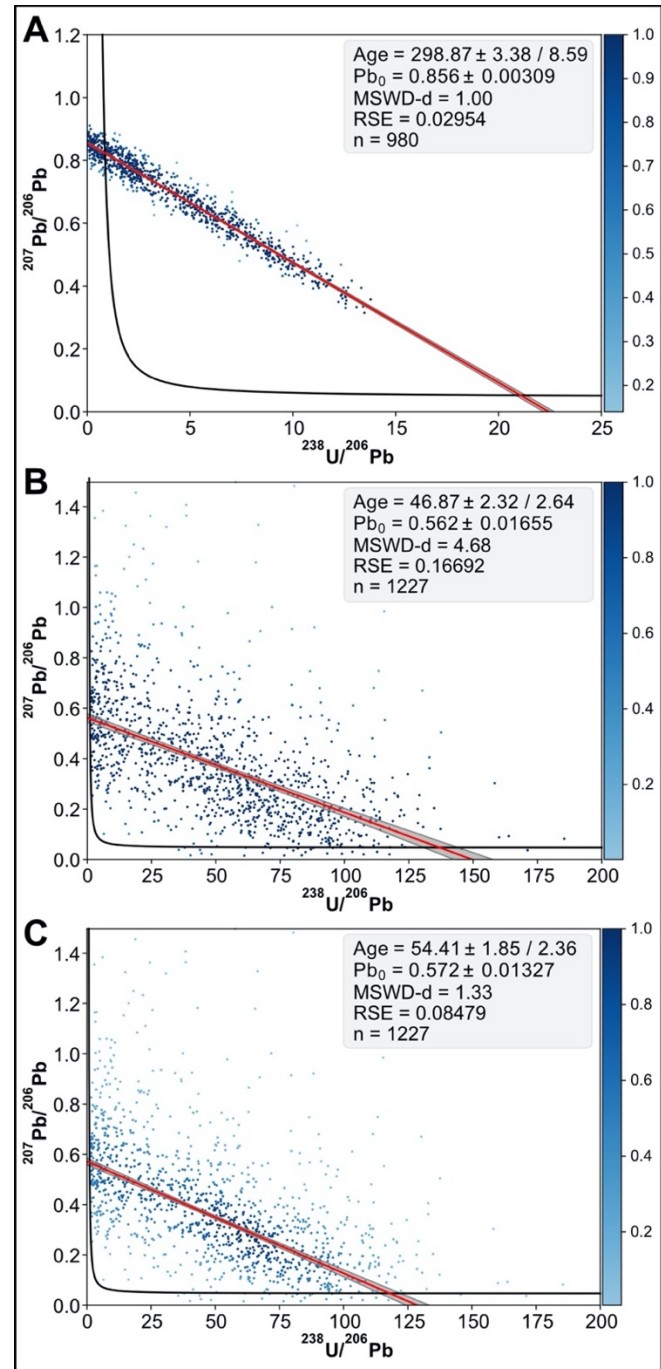

**Figure 6: Results obtained for samples JS4 (A) and BM18 (B, C). A: TW concordia plot calculated by robust regression for sample JS4. B, C: TW concordia plots calculated by robust regression for sample BM18. Pixels with $^{238}U/^{206}Pb$ values higher than 200 are not shown (n = 4). In C, the robust regression is weighted, with pixel weights calculated from a 2D Kernel Density Estimate (see text for details).**

660

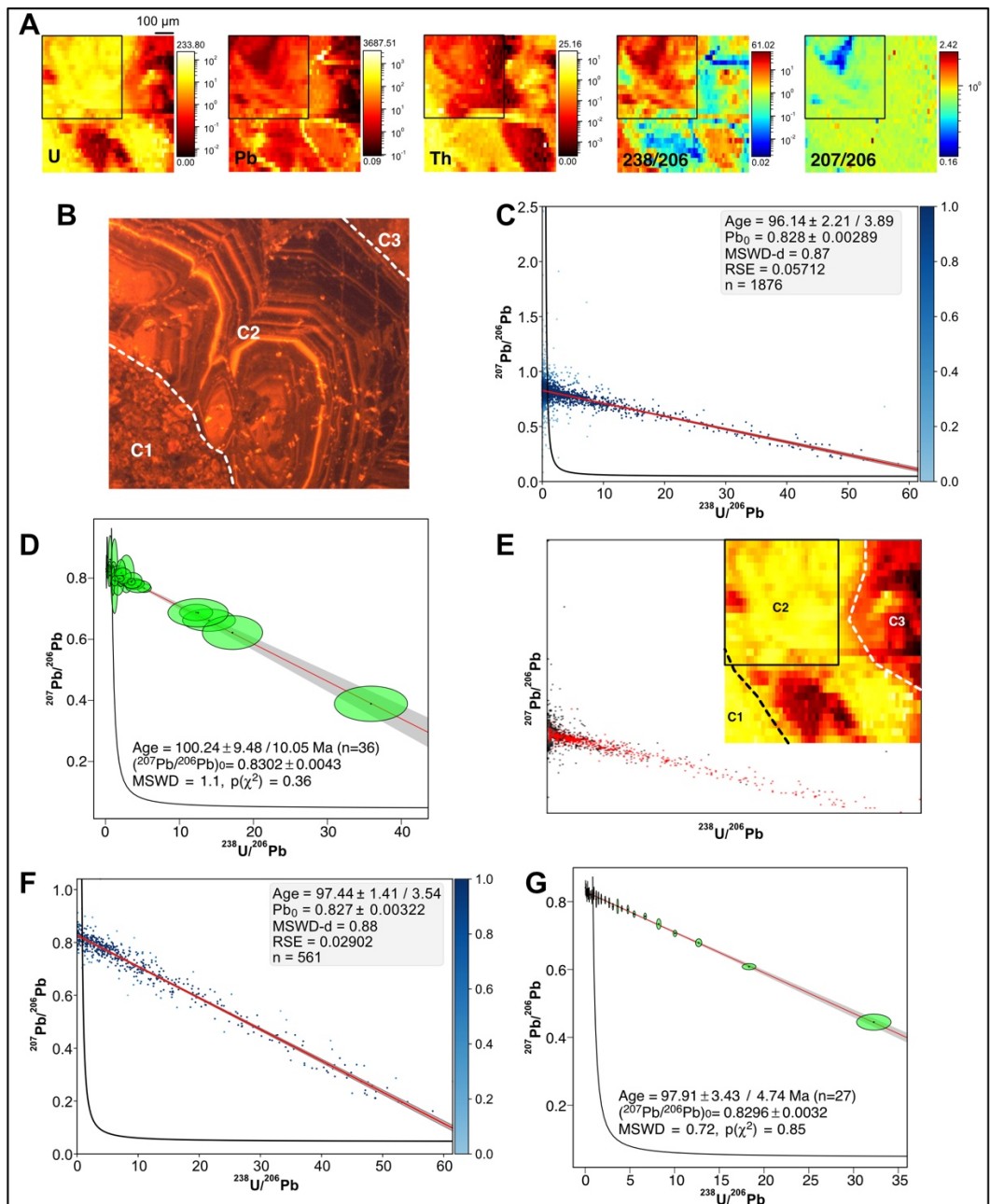

**Figure 7: Results obtained for sample ETC2. A:** Semi-quantitative element concentrations in ppm (U, Pb, Th) and $^{238}U/^{206}Pb$, $^{207}Pb/^{206}Pb$ isotopic maps. Black frames correspond to the image subset selected by colocalization. **B:** Cathodoluminescence image of an area close to that of the image map. Three distinct cements are distinguished (C1 to C3). **C:** TW concordia plot with ages and their uncertainty calculated by robust regression for the entire map. **D:** TW concordia plot calculated from discretization of the entire isotopic map ('pseudo-spots'). **E:** $^{207}Pb/^{206}Pb$ versus $^{238}U/^{206}Pb$ scatterplot obtained with the ScatterJn plugin from the isotopic maps. Red dots correspond to the pixels of the image subset (black frame in A and on the Uranium concentration map where the approximate location of distinct cement is given). **F:** TW concordia plot with ages and their uncertainty calculated by robust regression for the image subset. **G:** TW concordia plot obtained with the approach of Drost et al. (2018) for the entire isotopic map.

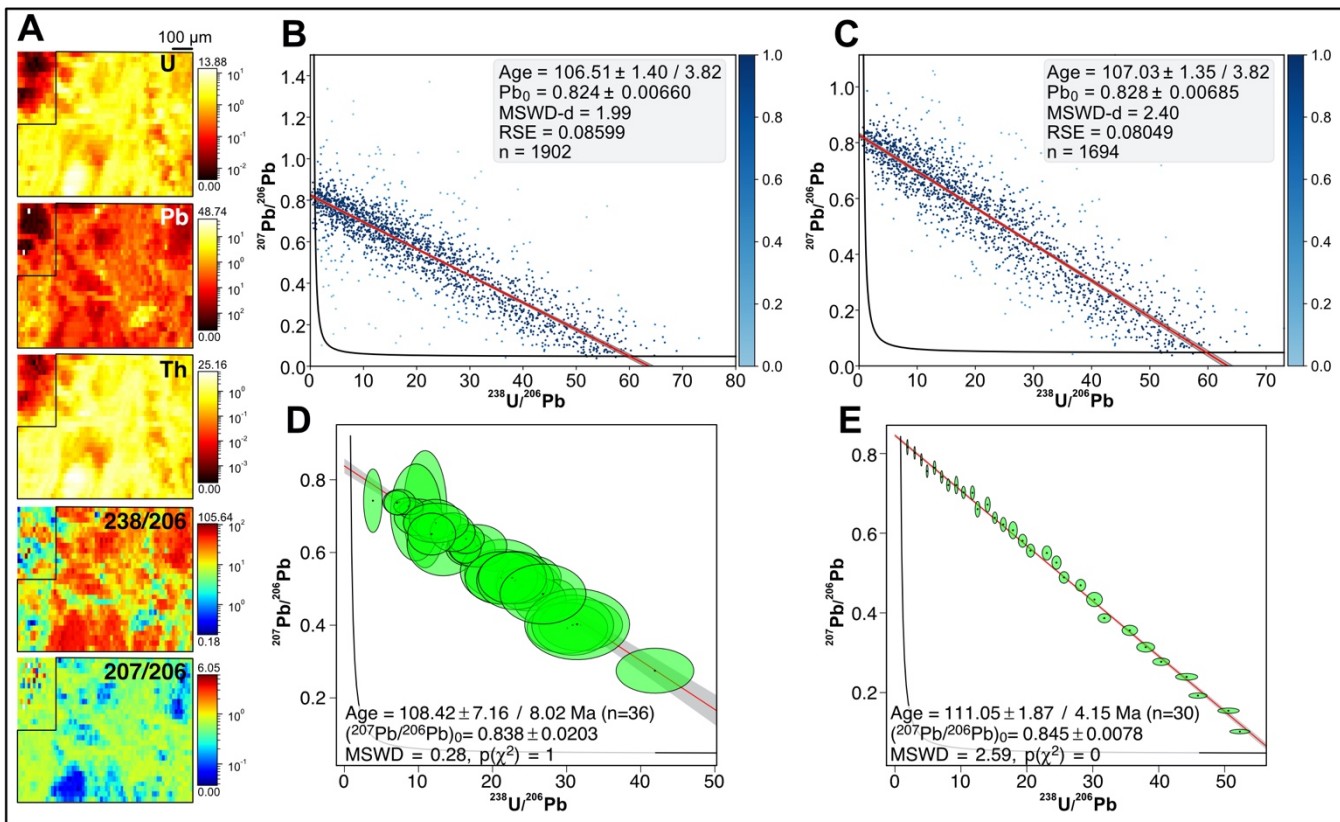

Figure 8: Results obtained for sample ARB. A: Semi-quantitative element concentrations in ppm (U, Pb, Th) and $^{238}U/^{206}Pb$, $^{207}Pb/^{206}Pb$ isotopic maps. B: TW concordia plot with ages and their uncertainty calculated by robust regression. Pixels with $^{238}U/^{206}Pb$ and $^{207}Pb/^{206}Pb$ values above 80 and 1.4, respectively, are not shown (n = 11). C: Same as B, except that the pixels outside the black frames on the isotopic maps (A) were not considered in age calculation, based on the colocalization study. D: TW concordia plot calculated from discretization of the entire isotopic map ('pseudo-spots'). G: TW concordia plot obtained with the approach of Drost et al. (2018) for the entire isotopic map.

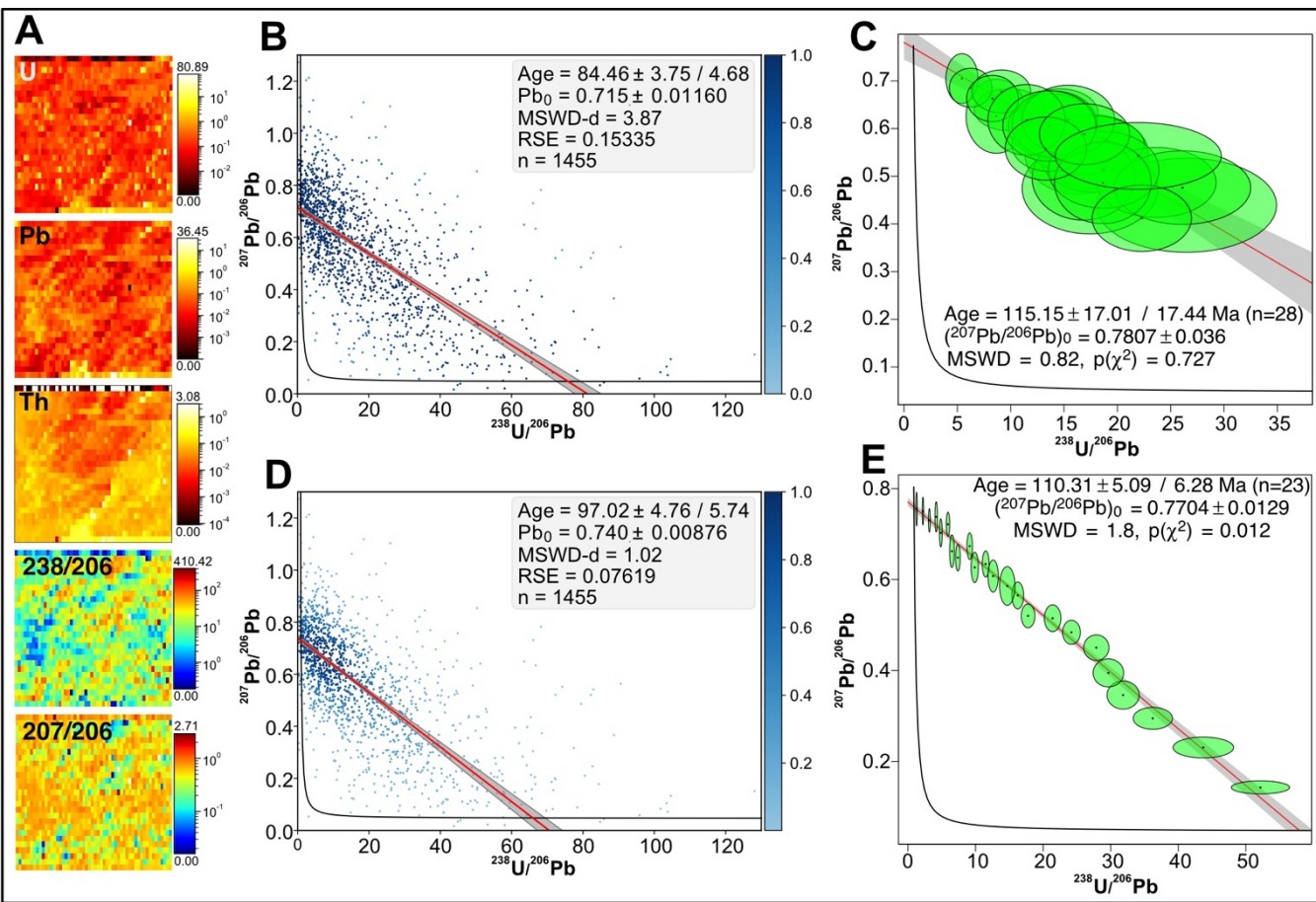

**Figure 9: Results obtained for sample C6-265-D5. A: Semi-quantitative element concentrations in ppm (U, Pb, Th) and $^{238}U/^{206}Pb$, $^{207}Pb/^{206}Pb$ isotopic maps. B: TW concordia plot with ages and their uncertainty calculated by robust regression. Pixels with $^{238}U/^{206}Pb$ and $^{207}Pb/^{206}Pb$ values above 130 and 1.3, respectively, are not shown (n = 13). C: TW concordia plot calculated from discretization of the isotopic map ('pseudo-spots'). D: TW concordia plots calculated by weighted robust regression, with pixel weights calculated from a 2D Kernel Density Estimate. E: TW concordia plot obtained with the approach of Drost et al. (2018).**

685