# Peer review of "Direct U-Pb dating of carbonates from micron scale fsLA-ICP-MS images using robust regression"

_Geochronology, 2020_

## Referee Comment (RC1) · Anonymous Referee #1 · 27 May 2020

The paper by Hoareau et al. presents a new approach to U-Pb LA-ICP-MS dating of carbonates that uses a mapping strategy in combination with robust regression of pixel U-Pb ratios. The authors suggest the direct plotting of pixel values into isochron diagrams as a simple approach that allows easy visual access of data quality and can be applied in combination with pixel filtering or pixel colocalization. Image-guided or image based approaches to LA-ICP-MS dating of carbonate clearly have the advantage that some certainty about genetic homogeneity (or heterogeneity) of a sample can be gained. However, to collect a set of masses that are indicative of inclusions, alteration, or different generations of carbonate minerals in general requires a fast-scanning ICP-MS instrument. The analytical setup used to acquire the data for the submitted study

is not capable of fast mass scanning and thus no chemical information in addition to U-Th-Pb isotope data was collected. In order to still test for age homogeneity of the mapped portion of the sample, a pixel value colocalization approach is used. This allows to obtain information on the location and distribution of pixels representing outliers identified by plotting U-Pb data in isochron diagrams and to draw subsets of the data for testing equivalence of the results.

As advance in instrumentation and software allows or even requires exploring new strategies for data acquisition and processing, the presented approach seems to be a good addition to the methods currently employed. I am, however, not a statistician and can therefore not judge whether the statistical base is appropriately chosen and correctly applied to the presented workflow. The manuscript is mostly clearly structured and written but needs clarification in several points. The English language may need some polishing too.

In the current version of the manuscript there is a lack of clarity on (1) how (or if) the data processing protocol deals with correction for instrumental drift, (2) how many NIST612 analyses are used for pixel ratio normalization (section 2.3.2; just the NIST analyses directly preceding and following that of the respective sample or those of the entire analytical session? And if the latter whether or not any excess scatter of the background- [and drift-?] corrected isotope ratios was observed) and (3) how the correction for radiogenic $208Pb$ from in-situ Th decay in Th-bearing samples is applied for the use of common $208Pb$ in the respective isochron plots. This should be clarified.

The identification of outlier pixel locations by colocalization is an attempt to data filtering that is not always efficient or successful. Apart from the usual "stray values" it may detect areas with low U and/or Pb concentrations where large scatter can exist due to insufficient counting statistics. Such scatter is inherent at very low ion signals and does not always justify exclusion of the data (key words: limit of detection, limit of quantification). The colocalization may also be successful in identifying areas with U loss (when surrounded by areas with pristine U-Pb systematics), where highly radiogenic
Pb isotope compositions at low 238U/206Pb would occur. The colocalization technique is, however, likely to fail in a case where different types/generations of carbonate minerals yield data point arrays that are intersecting each other in a small angle resulting in largely overlapping data point arrays. Please see comments to sections 3.1.4 and 3.2.2 for further explanation.

The lack of reported (numerical) data is also an issue. According to their data policy the journal requires a statement on where the data underlying a study can be found. Although the study presents/uses new data, there are no data submitted with the ms and a download link or a statement on data availability are missing. The approach presented in the study is new and associated data tables would be huge when choosing a format that would allow the reviewer/reader to plot and play with the data. Yet for potential further review and the case the manuscript is accepted for publication the authors should come up with an idea on data reporting and make the data accessible (Raw data files? Fully processed numerical pixel data? At the very least fully processed numerical values for sets obtained by discretization?...).

Specific comments

Abstract: Reference citations are normally avoided in the abstract.

1. Introduction:

first paragraph - the authors may want to clarify that the very good spatial resolution allows to detect and exploit sub-millimetre scale heterogeneities in U and initial Pb concentration

second paragraph - "...spot ablations of sizes close to 100 $\mu$m..." this statement is too imprecise; my own quick literature survey revealed that spot diameters range at least from 80 to 235$\mu$m depending on lab and U concentration of the samples. The actual range of spot diameters should be used here and a few citations can be added to give somewhat more credit to previous work in LA-ICPMS U-Pb carbonate dating.

third paragraph (from line 48) - it is not only detrital contamination that can be revealed by additional minor and trace element mapping but also alteration zones and different types or generations of carbonate minerals

2.2.2 from Line 116: (i) lines of 50$\mu$m width separated by 100$\mu$m distance - this is confusing as it would leave 50 or 150$\mu$m wide gaps between lines but Fig. S2 does not suggest there are gaps. Please rephrase. from Line 123: (ii) "...lines of 25$\mu$m width, separated by a distance of 25$\mu$m..." - again is this the spacing required by the "spot size" or are there 25$\mu$m gaps between the lines that are not ablated (Fig. 1 suggests the former) from Line 128: is the preablation (and second map) run directly over the previous screening map or is the sample repolished between screening and dating experiments? Line 130: the reference age of WC-1 is here given with 254.4 $\pm$ 7 Ma while Table 7.1 quotes 254.4 $\pm$ 6 Ma. Why not just use the original age of 254.4$\pm$6.4 Ma given by Roberts et al. (2017) for WC-1.

2.3.2 Lines 148/149 " The standard error (i.e., standard deviation of the mean) for each ratio was also calculated from the standard deviation which is an output of the function." - is this referring to NIST612 analyses? Line 153 - was an uncertainty applied when anchoring the regression? Roberts et al. (2017) give 0.85$\pm$0.04 for WC-1.

2.3.4 last sentence: The long-term excess variance cannot be derived from the primary reference material. Instead a quality control material (such as Duff Brown) should be used to assess this parameter.

2.3.5 - Residual Standard Error - A short statement on the range of acceptable RSE for different signal intensities could be given as dispersion is not only ruled by age homogeneity of the sample but also by counting statistics. - discretizing of pixels into sets by increasing 238U/206Pb ratios - what are acceptable values for d-MSWD? And wouldn't it be more appropriate to group the pixels into sets along the regression line rather than along the x-axis (i.e. 238U/206Pb) as the data points are scattered in both the x- and y-directions

2.3.6 Lines 209/210: "...not necessarily more accurate..." - shouldn't this read "not necessarily more precise"?

3.1 "... This is due to Pb0 values lower than that calculated from data of Hill et al. (2016) (0.738 ± 0.01, 1s). The discrepancy between both values is explained by the high 238U/206Pb ratio of the sample (typically 60-70) that prevents a precise determination of the common lead value. ..." It is not the high 238U/206Pb ratio or the low 207Pb/206Pb intercept itself but rather the lack of spread of the data points along a linear array which results in a poorly defined slope of the regression line and thus in imprecise and inaccurate values for both intercepts. Please clarify. Also an uncertainty for the initial 207Pb/206Pb composition of Duff Brown is quoted but anchoring is done without uncertainty as suggested by Fig. 2.

3.1.2 Line 227: "...large variation in the U/Pb ratios and a large amount of U..." - both parameters could be specified by giving the approx. range of U/Pb ratios and an approximate U concentration or the range of U concentration in brackets Lines 233-235:" This high precision can be related to the very good visual and statistical parameters calculated for this regression (good point alignment, very low RSE, and low d-MSWD..." - again the values could be given here rather than the narrative description of these values (also applies to further sections)

3.1.4 The pixel data show a lot more scatter around the regression line than the error ellipses of the spot data, i.e. the band in which the pixel data fall is much wider than a band covering the scattered (!) ellipses. This is not surprising but may be a problem. While spots were likely placed in a way not overlapping different textures (and thus error ellipses reflect the heterogeneity of the sample), the maps include all textural components and the pixel data represent a mixture of these different components. If there are small variations in ages and/or common Pb compositions for the different textural components this would not be obvious from plotting the pixel data as the different data point arrays are largely overlapping and may be intersecting each other in a small angle. The colocalization can be useful to locate areas that are rather heavily

affected by for instance U loss but will not necessarily be able to detect subtle differences in U-Pb systematics that result in slightly different slopes and intercepts of the regression lines of different components within the sample. One possibility to test for homogeneity might be dividing the map in squares/tiles with a size of a certain number of pixels (e.g., 125 x 130$\mu$m or corresponding to 20s) and to then check the alignment of the resulting ellipses. This is just one thing that could be tried but I don't know if this is feasible with the software used.

3.2.1 Line 284: "more precise" rather than "more accurate"? It looks like about half of the data for cement C2 are squandered and not used for age calculation when selecting a rectangular area...

3.2.2 From the description in the text and the caption of Figure 7 it is not clear to me which pixels were actually used for age calculation. The outlined area in the top left corner of the maps in Figure 7 does however seem to have very low U and Pb concentrations and larger deviations of such data points from the regression line can be expected just from counting statistics.

There seems to be an accident with the numbering of the sections as 3.2.2 is followed by 5.

5.1 Lines 309/310: "If not, the operator must remain attentive to the quality of the data, by checking the scattering of the pixels around the regression lines and the possible presence of phase mixtures." - this may not be enough (see comment to section 3.1.4) Lines 311/312: "The use of a colocalization approach as presented in our study is an efficient way to carry out this careful study of the spatial distribution of the furthest pixel values from the regression line." - see comments to sections 3.1.4 and 3.2.2. The values furthest away from the regression line are likely not too indicative of sample homogeneity/heterogeneity but often represent "true outliers" that cannot be related to age homogeneity in the sample. Line 322: "... maps can be built directly on thin sections (work of Drost et al., 2018) or on thick sections (this work)..." - The use of

thin sections (in the actual meaning, i.e. $\sim25\mu$m thickness) for high repetition rate mapping experiments in carbonates is dangerous due to preparation-related potential local deviations from the targeted thickness of the thin section and due to potentially heterogeneous sample material that may result in different local ablation rates. The experiments of the presented study resulted in 25 to 35$\mu$m depths confirming this and Drost et al. 2018 state the use of a "...polished rock slab of the sample..." instead of a thin section. The sentence should be rephrased avoiding to encourage the use of thin sections.

7.1 Analytical conditions table: There is a comment that "Ages in the data table are quoted at 95% absolute ..." I could not find a data table in the manuscript or supplement.

Figure captions (Figs. 2 - 5, 7): is "pseudo-ellipses" the intended term?

Figures: Fonts in figures appear to be too small.

Figure S2: which element or isotope ratio is shown on the maps?

---

## Referee Comment (RC2) · Anonymous Referee #2 · 27 May 2020

This is an interesting manuscript that shows the first application of a fs LA system to U-Pb dating of carbonates. While a method using an imaging approach similar as in Drost 2018 is used, a direct comparison including differences and improvements is only partly given and should be improved. To show that the robust regression works similar or better than the approach by Drost 2018, a direct comparison on the same data set using both approaches should be given. Generally, the improvements and new findings compared to existing and published methods should be more emphasised, including the use of fs LA, its possibility of high repetition rates, fast scanning, and the ease of use of the robust regression of individual points. Please provide the "raw" data of your images so that the interested scientist can look and play with the data

themselves. Please be consistent and always us ICP-MS or ICPMS. Abstract: Quite a few carbonate ages are published using quadrupole ICP-MS, I suggest to generally talk about ICP-MS in the abstract. Line 57-58:" Additional examples of the interest of this new approach are provided in Roberts et al. (2019)." This sentence does not fit here.

Line 87: Please clarify where the age with poor statistics is coming from.

Line 105: mixing of He aerosol flow with Ar "in" the ICP-MS ? Please be more precise.

Line 110: I do not think it is relevant that the ICP-MS used is a HR instrument, but a sector field.

Line 116: The First image: No pre-cleaning pulses? How do you recognise surface Pb contamination?

Line 132: If you apply the robust regression that puts the lowest weight to the outliers, why is in your procedure a second step necessary rejecting 2.5% outliers? What is the difference to the results without rejection?

Line 141: Each pixel of the image consists of 8 measurements (average) it should then be possible to calculate an uncertainty, and a different regression approach might be possible for comparison with the robust regression presented here.

Do you do the first and second image on the same day, same sequence? How long does it take to analyse image 1 get the image as presented in Figure S2? What is the criterion to select the region for image 2? (I would guess highest U/Pb variability or highest U concentration). Based on Figure S2 this is not clear or rather random especially for Sample ETC2 as the image 2 is outside image 1, Why?

Drift correction with RM measured only every 38 to 76 minutes? No Drift correction for the U/Pb ratio? Please describe your approach in drift correction and its influence on uncertainties in more detail.

All Figures with the robust regression have a white to blue, 0.x-1 colour scheme indicated on the right (I assume the weight as described in section 2.3.3?), but nowhere explained what it means. There are also open circle symbols likely the outliers that are not described. Please give this information either directly in the figures, or the figure caption, or leave it and just have points. Light blue to white points are hardly visible on white background.

Section 7.1 Please give the sensitivity of your instrument as % of # ions detected of # of atoms ablated, e.g. for a volume measured crater in NIST 610 and a measurement of U only.

Please mention what kind of ablation cell is used, single volume, 2 volume, manufacturer, size, shape etc.

What is the possible sample throughput of your system per day with the described method?

Out of curiosity, what kind of cones (Jet sampler and H or X skimmer) do you use in combination with the Jet Interface of your Element XR? (this does not need to be part of the manuscript)

Figure S2: mn should be min. Please indicate what is plotted either in the figure itself or the caption (238U/206Pb)?

---

## Author Comment (AC1) · 9 Jul 2020

RC1: "The paper by Hoareau et al. presents a new approach to U-Pb LA-ICPMS dating of carbonates that uses a mapping strategy in combination with robust regression of pixel U-Pb ratios. The authors suggest the direct plotting of pixel values into isochron diagrams as a simple approach that allows easy visual access of data quality and can be applied in combination with pixel filtering or pixel colocalization. Image-guided or image based approaches to LA-ICP-MS dating of carbonate clearly have the advantage that some certainty about genetic homogeneity (or heterogeneity) of a sample can be gained. However, to collect a set of masses that are indicative of inclu-

sions, alteration, or different generations of carbonate minerals in general requires a fast-scanning ICPMS instrument. The analytical setup used to acquire the data for the submitted study is not capable of fast mass scanning and thus no chemical information in addition to U-Th-Pb isotope data was collected. In order to still test for age homogeneity of the mapped portion of the sample, a pixel value colocalization approach is used. This allows to obtain information on the location and distribution of pixels representing outliers identified by plotting U-Pb data in isochron diagrams and to draw subsets of the data for testing equivalence of the results".

Thank you for the positive comments. We clearly state in the initial manuscript that dating based on LA-ICP-MS isotope mapping is more suited to fast-scanning ICP-MS, so that the approach proposed by Drost et al. (2018) should be used (l. 247-248, for sample PXG20-1). Clearly, the colocalization study as we propose is an alternative way to select the pixels most amenable to precise dating. If fast-scanning ICPMS is available, we would suggest to first use chemical information to select a first area (Drost et al.), followed by our direct plotting of pixel values into diagrams and robust regression strategy to calculate the age.

RC1: "As advance in instrumentation and software allows or even requires exploring new strategies for data acquisition and processing, the presented approach seems to be a good addition to the methods currently employed. I am, however, not a statistician and can therefore not judge whether the statistical base is appropriately chosen and correctly applied to the presented workflow. The manuscript is mostly clearly structured and written but needs clarification in several points. The English language may need some polishing too."

Thank you for these mostly positive comments. The English language has been checked again.

RC1: "In the current version of the manuscript there is a lack of clarity on (1) how (or if) the data processing protocol deals with correction for instrumental drift, (2) how

many NIST612 analyses are used for pixel ratio normalization (section 2.3.2; just the NIST analyses directly preceding and following that of the respective sample or those of the entire analytical session? And if the latter whether or not any excess scatter of the background- [and drift-?] corrected isotope ratios was observed) and (3) how the correction for radiogenic 208Pb from in-situ Th decay in Th-bearing samples is applied for the use of common 208Pb in the respective isochron plots. This should be clarified."

The comments proposed here are justified. In its first version, the text does not allow for a precise estimate of the way the corrections of isotopic ratios were made. To answer points 1) and 2): only isotopic maps of reference materials NIST 612 and WC1 analyzed just before and after the unknown have been considered. For pixel Pb/Pb ratio of the unknown and of WC1, normalization (correction of mass bias) was performed using NIST 612 based on a standard bracketing approach, with the (robust) mean of NIST612 pixel values calculated as described in the text. The reference values of the NIST612 used for normalization for the lead ratios were taken from Woodhead and Hergt (2001), which was not specified in the main text. The drift-correction of the U/Pb ratios was also based on a standard bracketing approach, with (robust) mean U/Pb value of the NIST612 preceding the unknown. This choice is the result of constraints related to the equipment available at the time of the study. Our home-made ablation cell cannot accommodate several samples simultaneously (unknowns + RMs) as it is commonly the case for commercial setups used in geochronology. Therefore, repeated RM analysis along with an unknown sample, and analysis of several unknowns (and associated RMs) during one session (for example, half a day) requires time-consuming sample handling and cannot be automated. As a result, our analysis workflow does not allow the calculation of an excess scatter by the repeated analysis of a homogeneous secondary standard, or by a pseudo-secondary standard approach based on Paton et al (2010) and as implemented in Iolite©. Ideally, when future equipment will be available, we will move to a more traditional data reduction workflow based on repeated RM analysis, and the use of a dedicated software. To answer point 3): we agree

that this point has not been addressed in the manuscript, and will be added in the revised version. The followed method was similar to that proposed by Parrish et al. (2018), except that the ratios were calculated for each pixel rather than for individual spot analyses. The amount of radiogenic 208Pb was calculated from measured 232Th and the age given by the T-W regression, allowing estimation of the amount of common 208Pb (208Pbc). The 208Pbc/206Pb, 206Pb/208Pbc, and 238U/208Pbc ratios are then straightforward to calculate. Given the low decay ratio of 232Th, and the young age of most analyzed samples, variations in the value of the age chosen for calculation of radiogenic 208Pb of up to 10 Ma have negligible impact on the final age.

RC1: "The identification of outlier pixel locations by colocalization is an attempt to data filtering that is not always efficient or successful. Apart from the usual "stray values" it may detect areas with low U and/or Pb concentrations where large scatter can exist due to insufficient counting statistics. Such scatter is inherent at very low ion signals and does not always justify exclusion of the data"

Indeed, a pixel filtering approach, whatever it is (trace element concentrations, isotope ratios, counting statistics, or colocalization) is not always perfectly efficient or successful. In the proposed approach, pixel selection based on colocalization in Tera-Wasserberg type diagrams must be seen as a guide to potentially use a spatial subset of the isotopic maps to get more precise (and hopefully more accurate) ages. This tool has clear limitations in the sense that it requires spending time analyzing isotope ratio images and it does not allow judging the origin of a possible dispersion of pixel values. In one of our examples (ARB), such scatter is indeed linked to a low U concentration. Is the rejection of the corresponding area justified? We note that the filtering of pixels based on rejection of low counts for 238U (among other filtering approaches) has been proposed in the referential publication of Roberts et al. (2019), for the image-based dating of calcite sample NR1511. However, in our approach we do not reject pixels potentially distributed on several portions of the map; we select instead a map of lower area than the initial one, for which calculated ages are similar within the uncertainties,

but more precise and with better statistics, than if the entire map is considered. This can be compared to the practice in LA-ICP-MS spot dating of carrying out several transects across a sample, and selecting the one giving the best results. An example is provided in Roberts et al. (2019) (their figure 7a), where the image-based choice of the transect is guided by higher U concentration. This is equivalent to the rejection of transects corresponding to the lowest U concentrations. finally, as stated several times in the manuscript (see above), colocalization was used because filtering based on trace element concentrations could not be performed with the available analytical set-up. We do not advocate the use of colocalization tooth and nail, but we suggest that it can be an interesting alternative (or better a complement) to the filtering approaches proposed elsewhere.

RC1: " (key words: limit of detection, limit of quantification)."

Based on NIST612, the limit of detection as of October 2019 was ca. 1.3 ppb and 0.2 ppb for $206Pb$ and $238U$, respectively. The limit of quantification was ca. 4.3 ppb and 0.8 ppb for $206Pb$ and $238U$, respectively. These values will be added to the revised version of the manuscript.

RC1: "The colocalization may also be successful in identifying areas with U loss (when surrounded by areas with pristine U-Pb systematics), where highly radiogenic Pb isotope compositions at low $238U/206Pb$ would occur. The colocalization technique is, however, likely to fail in a case where different types/generations of carbonate minerals yield data point arrays that are intersecting each other in a small angle resulting in largely overlapping data point arrays. Please see comments to sections 3.1.4 and 3.2.2 for further explanation."

We agree with this comment. See our answer below (dealing with sections 3.1.4 and 3.2.2)

RC1: "The lack of reported (numerical) data is also an issue. According to their data policy the journal requires a statement on where the data underlying a study can be

Interactive
comment

found. Although the study presents/uses new data, there are no data submitted with the ms and a download link or a statement on data availability are missing. The approach presented in the study is new and associated data tables would be huge when choosing a format that would allow the reviewer/reader to plot and play with the data. Yet for potential further review and the case the manuscript is accepted for publication the authors should come up with an idea on data reporting and make the data accessible (Raw data files? Fully processed numerical pixel data? At the very least fully processed numerical values for sets obtained by discretization?...)."

We propose to add raw normalized (Pb/Pb ratios) and drift-corrected (U/Pb) pixel data as supplementary material.

RC1: "Specific comments

Abstract: Reference citations are normally avoided in the abstract."

The abstract will be modified accordingly.

RC1: "1. Introduction: first paragraph - the authors may want to clarify that the very good spatial resolution allows to detect and exploit sub-millimetre scale heterogeneities in U and initial Pb concentration"

We agree. It will be clarified.

RC1: "second paragraph - "...spot ablations of sizes close to 100 $\mu$m..." this statement is too imprecise; my own quick literature survey revealed that spot diameters range at least from 80 to 235 $\mu$m depending on lab and U concentration of the samples. The actual range of spot diameters should be used here and a few citations can be added to give somewhat more credit to previous work in LA-ICPMS U-Pb carbonate dating."

We agree, and propose to modify the sentence as follows: "Most carbonate dating LA-ICP-MS studies are based on spot ablations of sizes ranging from 80 to 235 $\mu$m (e.g., Roberts and Walker, 2016; Parrish et al., 2018; Ring and Gerdes, 2016)"

RC1: "third paragraph (from line 48) - it is not only detrital contamination that can be revealed by additional minor and trace element mapping but also alteration zones and different types or generations of carbonate minerals"

We agree, and propose to modify the sentence as follows: "However, in this case, the concentrations of elements characteristic of detrital contamination (e.g., Zr, Rb etc), or allowing to locate alteration zones or distinct calcite cement generation (e.g., U, Pb, Fe, Mn) are used to define pixel exclusion thresholds. These are used to filter out the values corresponding to microscopic inclusions of other minerals (e.g., clays) and altered zones, and/or to isolate the cement generation targeted for dating"

RC1: "2.2.2 from Line 116: (i) lines of 50 $\mu$m width separated by 100 $\mu$m distance - this is confusing as it would leave 50 or 150 $\mu$m wide gaps between lines but fig. S2 does not suggest there are gaps. Please rephrase."

To be more precise, the lines of 50 $\mu$m height (not width) have been made every 100 $\mu$m, leaving a gap of 50 $\mu$m between them. On the preliminary isotopic maps of fig. S2, the lines corresponding to these gaps have been removed from the image to avoid a zebra-type rendering which made visualization difficult. Since this would result in images of height divided by 2, the original aspect ratio was restored by stretching the image back to its original shape. Logically, the final pixels are therefore 100 $\mu$m height instead of 50 $\mu$m. This will be specified in the figure caption.

RC1: "from Line 123: (ii) "...lines of 25 $\mu$m width, separated by a distance of 25 $\mu$m..." - again is this the spacing required by the "spot size" or are there 25 $\mu$m gaps between the lines that are not ablated (fig. 1 suggests the former)"

The lines of 25 $\mu$m height have been made every 25 $\mu$m, so that they are immediately adjacent to each other.

RC1: "from Line 128: is the preablation (and second map) run directly over the previous screening map or is the sample repolished between screening and dating experiments?"

The sample is repolished between the maps.

RC1: "Line 130: the reference age of WC-1 is here given with 254.4 ± 7 Ma while Table 7.1 quotes 254.4 ± 6 Ma. Why not just use the original age of 254.4±6.4 Ma given by Roberts et al. (2017) for WC-1."

We agree, it will be corrected.

RC1: "2.3.2 Lines 148/149 " The standard error (i.e., standard deviation of the mean) for each ratio was also calculated from the standard deviation which is an output of the function." - is this referring to NIST612 analyses?"

Yes. It will be reminded in the revised manuscript.

RC1: "Line 153 - was an uncertainty applied when anchoring the regression? Roberts et al. (2017) give 0.85±0.04 for WC-1."

There are currently no implementations of anchored regressions with uncertainties on the y-intercept in the R language, which is used for robust regressions in our study. The same limitation arises when using IsoplotR (Vermeesch, 2018) for anchored York-type regressions.

RC1: "2.3.4 last sentence: The long-term excess variance cannot be derived from the primary reference material. Instead a quality control material (such as Duff Brown) should be used to assess this parameter."

Of course. It will be corrected.

RC1: "2.3.5 - Residual Standard Error - A short statement on the range of acceptable RSE for different signal intensities could be given as dispersion is not only ruled by age homogeneity of the sample but also by counting statistics. "

Concerning the Residual Standard Error, you raised a relevant question. Making the

link between signal intensity and RSE would necessitate several tests on appropriate carbonate standards of known composition, with varying laser ablation intensity. This will be hopefully done in future analytical sessions. Our experience with analyzed carbonate samples is that successful samples all had an RSE lower than ca. 10-12% of the y-intercept value (i.e., < 0.1 for a common Pb value of 0.85 in Tera-Wasserburg diagrams), whatever the average concentrations in U and Pb. Note that the RSE allows a quick estimation of the scatter around the regression line (residuals), but good overall alignment of the points, which is a better estimate of age homogeneity, is more conveniently estimated with the d-MSWD and the running mean. Indeed, LA-ICP-MS spot analyses demonstrate that even scattered data can lead to meaningful ages, as soon as the MSWD reaches acceptable values.

RC1: "- discretizing of pixels into sets by increasing 238U/206Pb ratios - what are acceptable values for d-MSWD?"

A perfect alignment of the ellipses produced from the pixel value dataset will give a value of 1 for the d-MSWD, as for the MSWD in York-type regressions. Our empirical experience on analyzed samples shows that meaningful ages are obtained with d-MSWD lower than about 2.5, which is higher that the theoretical expectations for the critical MSWD values in York-type regressions, for more than 6 degrees of freedom (Wendt and Carl, 1991). However, contrary to the latter, there is currently no theoretical basis for the definition of isochrons versus errorchrons based on the d-MSWD, as reminded in section 5.2. We are currently working on a robust goodness-of-fit estimator for the robust regression approach, inspired by Powell et al. (2020).

RC1: "And wouldn't it be more appropriate to group the pixels into sets along the regression line rather than along the x-axis (i.e. 238U/206Pb) as the data points are scattered in both the x- and y-directions"

We agree and have changed the way of discretizing the pixel values. Note that the regression slopes are always close to 0 in the diagrams used for age calculation, which

will not noticeably change the results compared to the former method.

RC1: "2.3.6 Lines 209/210: "...not necessarily more accurate..." - shouldn't this read "not necessarily more precise"?"

Yes. The mistake will be corrected.

RC1: "3.1 "... This is due to Pb0 values lower than that calculated from data of Hill et al. (2016) (0.738 ± 0.01, 1s). The discrepancy between both values is explained by the high 238U/206Pb ratio of the sample (typically 60-70) that prevents a precise determination of the common lead value. ..." It is not the high 238U/206Pb ratio or the low 207Pb/206Pb intercept itself but rather the lack of spread of the data points along a linear array which results in a poorly defined slope of the regression line and thus in imprecise and inaccurate values for both intercepts. Please clarify."

You are entirely right. This is what we wanted to explain, but obviously it wasn't written clearly.

RC1: "Also an uncertainty for the initial 207Pb/206Pb composition of Duff Brown is quoted but anchoring is done without uncertainty as suggested by fig. 2."

See our answer concerning the common lead value of WC1.

RC1: "3.1.2 Line 227: "...large variation in the U/Pb ratios and a large amount of U..." - both parameters could be specified by giving the approx. range of U/Pb ratios and an approximate U concentration or the range of U concentration in brackets Lines 233-235:" This high precision can be related to the very good visual and statistical parameters calculated for this regression (good point alignment, very low RSE, and low d-MSWD..." - again the values could be given here rather than the narrative description of these values (also applies to further sections)"

The numerical values will be added to the revised version.

RC1: "3.1.4 The pixel data show a lot more scatter around the regression line than

the error ellipses of the spot data, i.e. the band in which the pixel data fall is much wider than a band covering the scattered (!) ellipses. This is not surprising but may be a problem. While spots were likely placed in a way not overlapping different textures (and thus error ellipses reflect the heterogeneity of the sample), the maps include all textural components and the pixel data represent a mixture of these different components. If there are small variations in ages and/or common Pb compositions for the different textural components this would not be obvious from plotting the pixel data as the different data point arrays are largely overlapping and may be intersecting each other in a small angle. The colocalization can be useful to locate areas that are rather heavily affected by for instance U loss but will not necessarily be able to detect subtle differences in U-Pb systematics that result in slightly different slopes and intercepts of the regression lines of different components within the sample. One possibility to test for homogeneity might be dividing the map in squares/tiles with a size of a certain number of pixels (e.g., 125 x 130 $\mu$m or corresponding to 20 s) and to then check the alignment of the resulting ellipses. This is just one thing that could be tried but I don't know if this is feasible with the software used."

We agree with the comment that the dispersion of pixel values is a problem, as indeed several generations of calcites with similar ages and/or common lead values could be erroneously confused. This strengthens the interest of a filtering approach as proposed by Drost et al (2018), which allows to clearly identify these generations on the basis of their composition. In that sense our colocalization can bring some help too. We note, however, that with a spot approach the problem raised is also possible; you write "While spots were likely placed in a way not overlapping different textures (and thus error ellipses reflect the heterogeneity of the sample)". All the problem lies in the "likely". It is also quite possible to mix phases with spots if an initial petrographic and geochemical study is not done properly, as for example two cements could be analyzed as one giving intermediate values. Nevertheless, we have implemented the requested test ("dividing the map in squares/tiles with a size of a certain number of pixels (e.g., 125 x 130 $\mu$m or corresponding to 20s)") in our code to check for heterogeneity. This
additional test will be now performed on a systematic manner. In the present study, the corresponding plots (represented in TW diagrams) will be added as supplementary data. The squares chosen have a dimension of 7 x 6 pixels, equivalent to 175 $\mu$m x 150 $\mu$m and 21 seconds of analysis, except for sample BH14 where they have a dimension of 4 x 6 pixels (100 $\mu$m x 150 $\mu$m; corresponding to 24 s). Although using such low number of pixels for mean and standard error calculations result in rather large uncertainties (i.e., large ellipses), the results are conclusive for all samples. The ellipses obtained are always well aligned and York-type regression in TW diagrams give the expected ages, but with uncertainties higher than with the robust method even without propagation of systematic uncertainty. The MSWD are always lower than 1, which is not surprising given the uncertainties of individual ellipses.

RC1: "3.2.1 Line 284: "more precise" rather than "more accurate"? It looks like about half of the data for cement C2 are squandered and not used for age calculation when selecting a rectangular area..."

We agree, the appropriate word is "precise". Concerning the chosen area, yes part of the isotopic map corresponding to cement C2 has not been used in this example. The idea was to show that even selecting a small part of the map could result in more precise ages than considering the entire map. Here, the subset was guided only by colocalization, rather than by petrographical analyses (and in the absence of trace element data, see lines 287-288). Combining both approaches shows that the subset belongs to cement C2 as identified from CL imaging.

RC1: "3.2.2 From the description in the text and the caption of figure 7 it is not clear to me which pixels were actually used for age calculation. The outlined area in the top left corner of the maps in figure 7 does however seem to have very low U and Pb concentrations and larger deviations of such data points from the regression line can be expected just from counting statistics."

We agree. The selected subset of the isotopic map corresponds to the entire map mi-

nus the top left corner. Although this selection was guided by colocalization, it appears that there is a strong link with U and Pb concentration, so scattered pixels are likely related with counting statistics.

RC1: "7.1 Analytical conditions table: There is a comment that "Ages in the data table are quoted at 95% absolute ..." I could not find a data table in the manuscript or supplement."

This is a mistake: ages in the figures, not in the data table.

RC1: "figure captions (figs. 2 - 5, 7): is "pseudo-ellipses" the intended term?"

We should have written "ellipses calculated from discretized pixel values".

RC1: "figures: Fonts in figures appear to be too small."

Their size will be increased as requested.

RC1: "figure S2: which element or isotope ratio is shown on the maps?"

These are the 238U/206Pb ratios. It will be specified in the caption.

---

## Author Comment (AC2) · 9 Jul 2020

RC2: "This is an interesting manuscript that shows the first application of a fs LA system to U-Pb dating of carbonates. While a method using an imaging approach similar as in Drost 2018 is used, a direct comparison including differences and improvements is only partly given and should be improved. To show that the robust regression works similar or better than the approach by Drost 2018, a direct comparison on the same data set using both approaches should be given."

Thank you for your comments. We have followed your recommendations by asking Kerstin Drost (which is warmly thanked) 2 sets of data corresponding to samples pre-

sented in Drost et al (2018) (JS4) and Roberts et al (2019) (BM18). Sample JS4 is characterized by high U concentrations (mean ca. 10 ppm) allowing precise ages to be obtained by the method of Drost et al. (2018) (297.8 ± 3.3 Ma, 297.2 ± 3.9 Ma, and 297.0 ± 2.9 Ma for the TW, 86TW and isochron diagrams, respectively). Those calculated with the robust regression method are similar to the previous within uncertainties (298.9 ± 3.4 Ma, 297.0 ± 3.4 Ma, and 300.8 ± 2.8 Ma for the TW, 86TW and isochron diagrams, respectively), with similar uncertainties and very good statistics (RSE < 5.5% of the y-intercept (see responses to the first reviewer), and d-MSWD ≤ 1). Note that to concur with results given by Drost et al. (2018), systematic uncertainties are not considered. Adding them leads to final age uncertainties above 8 Ma. For sample BM18, the low U and Pb concentrations (mean of ca. 200 pb and ca. 7 ppb, respectively) are logically associated with a large dispersion of U-Pb and Pb-Pb isotope ratios. The age obtained with the image-based approach as presented in Roberts et al (2019) (61.0 ± 1.7 Ma in the TW space) and with LA-ICP-MS spot analysis (59.5 ± 2.7 Ma; Beaudoin et al., 2018) could not be reproduced with our method. This is due to the presence of pixels with high $238U/206Pb$ acting as leverage points for the robust regression in the TW and 86TW diagrams. When systematic uncertainties similar to those of Beaudoin et al (2018) are considered, the ages are 46.9 ± 2.8 Ma and 48.4 ± 2.5 Ma for the TW and 86TW diagrams, respectively, more than 10 Ma younger than the expected one. For the isochron diagram, the age of 60.9 ± 2.1 Ma is similar to the expected age. Added to high RSE values (> 10% of the y-intercept value, see responses to the first reviewer), and d-MSWD values ranging from 2.5 to 4.9, these variable ages would lead us to consider this analysis as unreliable and to reject it. In order to improve our approach and get closer to the reference age, we have implemented the possibility to weight the pixel values used in the robust regression (weighted robust regression is made possible with the 'lmrob' library). Our choice is to attribute a weight to each pixel based on the density of pixels in the considered diagram. In the case of a large dispersion of data (especially related to counting statistics, even for a sample of homogeneous age), it is expected that the majority of points will still be clustered along the
isochron (i.e., higher density of points). Assigning more weight to these points should limit the impact of leverage points. Density in the 2D space (and so pixel weight) was estimated by a gaussian kernel density estimate (KDE), whose bandwidth is estimated by the Scott's Rule (Scott, 1992). Using this approach to the anomalous ages (TW and TW86) gives values similar within uncertainties to that of Beaudoin et al. (2018), but still centered towards younger ages ($54.4 \pm 2.6$ Ma and $55.0 \pm 2.5$ Ma for the TW and 86TW diagrams, respectively). The ages obtained with the 3 diagrams are not similar within uncertainty, which again would lead us to reject the analysis or, as they are close, to consider that the real age is likely comprised between 52 Ma and 63 Ma. We have tentatively reproduced the approach of Drost et al. (2018) by discretizing pixel ratio values and sorting them based on the 235/207 ratio (21 subsets). In the TW diagram, we obtain an age of $60.0 \pm 2.8$ Ma (MSWD = 0.95; with systematic uncertainties including long-term uncertainty), which is slightly younger than the value presented in Roberts et al. (2019). Making the same calculation for TW86 and isochron diagrams (not given in Roberts et al., 2019) gives values of $55.6 \pm 2.9$ Ma (MSWD = 1) and $60.1 \pm 6.4$ Ma (MSWD = 0.51), respectively. The 3 ages are centered between 55 Ma and 60 Ma, similar to those obtained with our approach. However, they are similar within uncertainties, which is not the case with the robust regression. Finally, we note that for the image-based dating obtained by Roberts et al. (2019) (TW diagram), the value of Pb0 (ca. 0.7) is significantly higher than that expected based on the spot analyses (ca. 0.59; Beaudoin et al., 2018)). The latter is, in contrast, almost identical to that obtained by weighted robust regression (ca. 0.57). For this sample of low U and Pb concentration, the approach of Drost et al. (2018) thus gives better results than the one presented in our study in terms of age, despite "wrong" y-intercept and slope values of the regression line (at least in the TW plot). We propose to discuss these differences in the revised manuscript, and to remind that for such kind of samples, spot-based dating should be preferred to image-based dating.

D.W. Scott, "Multivariate Density Estimation: Theory, Practice, and Visualization", John Wiley & Sons, New York, Chicester, 1992.

RC2: "Generally, the improvements and new findings compared to existing and published methods should be more emphasised, including the use of fs LA, its possibility of high repetition rates, fast scanning, and the ease of use of the robust regression of individual points."

The only studies devoted to U-Pb dating of carbonates from isotopic images are those of Drost et al (2018), Roberts et al (2019) and this work. The comparison between the approaches will be largely addressed in the revised version (see previous response). The advantages of the isotope imaging method compared to spot analyses will be discussed in more detail in section 5.1. While for the majority of the examples presented in our study (Duff Brown, BH14, PXG20-1, PXG32-2) we show that the ages obtained by imaging are identical with spot analyses (both in terms of value and uncertainty), the example of the BM18 sample, which will be added to the manuscript, shows that in the case of very low concentrations of U and Pb the spot approach seems to be more efficient. We have deliberately not deeply discussed the possible advantages / disadvantages related to the use of a high repetition rate fs laser, as this is not the aim of this study. The major advantage is the small beam size, which allows to build images of reduced size if necessary (pixels 12 x 25 $\mu$m), as already explained in the text. Instead, we wish to emphasize the ease of use of the robust regression dating approach, which can be used whatever the device used for image acquisition.

RC2: "Please provide the "raw" data of your images so that the interested scientist can look and play with the data themselves"

This will be done (see answer to R1)

RC2: "Please be consistent and always us ICP-MS or ICPMS. Abstract: Quite a few carbonate ages are published using quadrupole ICP-MS, I suggest to generally talk about ICP-MS in the abstract."

The required corrections will be done.

RC2: "Line 57-58:" Additional examples of the interest of this new approach are provided in Roberts et al. (2019)." This sentence does not fit here."

It will be removed.

RC2: "Line 87: Please clarify where the age with poor statistics is coming from."

The age was obtained in our laboratory by LA-ICP-MS spot analysis with the methodology detailed in the Supplementary material. It will be clarified in the text.

RC2: "Line 105: mixing of He aerosol flow with Ar "in" the ICP-MS ? Please be more precise."

The Argon, nitrogen and helium are all entering a twister spray chamber before reaching the plasma. This spray chamber has been modified by adding an additional inlet for the introduction of helium (transporting the ablated aerosol). It has been placed at the very top of the spray chamber and do not enter the chamber itself. We propose to add more detail to the text: "To improve sensitivity, 10 mL.min-1 of nitrogen was added to the twister spray chamber of the ICPMS via a tangential inlet while helium flow was introduced via another tangential inlet located at the very top of the spray chamber."

RC2: "Line 110: I do not think it is relevant that the ICP-MS used is a HR instrument, but a sector field."

OK.

RC2: "Line 116: The first image: No pre-cleaning pulses? How do you recognise surface Pb contamination?"

The images were done without pre-cleaning. Indeed, these images are only semi-quantitative, and aim to locate areas with both high U/Pb ratios and some spread in the ratios (see below).

RC2: "Line 132: If you apply the robust regression that puts the lowest weight to the outliers, why is in your procedure a second step necessary rejecting 2.5% outliers?

What is the difference to the results without rejection?"

We agree. This step is not really justified since the results obtained with and without outliers are almost similar. To follow your recommendations, we have decided to remove this step.

RC2: "Line 141: Each pixel of the image consists of 8 measurements (average) it should then be possible to calculate an uncertainty, and a different regression approach might be possible for comparison with the robust regression presented here."

The uncertainties (standard error) obtained on only 8 pixels are too high to make reliable York type regressions. Instead, we propose to perform regressions on averages and uncertainties calculated by separating the maps into several sub-maps, following the recommendations of the first reviewer. The ages obtained agree with those from the robust regression, but with much higher uncertainties.

RC2: "Do you do the first and second image on the same day, same sequence?"

The first image is used only as a guide to select the area most favorable for dating. They are usually not necessarily performed on the same sequence, neither on the same day.

RC2: "How long does it take to analyse image 1 get the image as presented in figure S2?"

We are not sure to understand the question. The time shown in figure S2 for each image corresponds to the analysis time required to obtain that image. The treatment to plot the image from raw data consists in a few lines of code, so it is done in less than a minute.

RC2: "What is the criterion to select the region for image 2? (I would guess highest U/Pb variability or highest U concentration). Based on figure S2 this is not clear or rather random especially for Sample ETC2 as the image 2 is outside image 1, Why?"

The criterion used are at the same time the presence of high U/Pb ratios and large variations in U/Pb ratios, on an area of size corresponding to images used for dating. This is not random. For sample ETC2, the wrong location is due to a mistake of the operator during the analytical session. We decided to present the map anyway, since despite this error the age obtained is satisfactory.

RC2: "Drift correction with RM measured only every 38 to 76 minutes? No Drift correction for the U/Pb ratio? Please describe your approach in drift correction and its influence on uncertainties in more detail."

There seems to be some confusion. The times shown in figure S2 correspond to the images used to identify areas suitable for dating. No standards are used to obtain these maps. Standards are used for the second maps, which are obtained in 19 to 38 minutes. For the latter, yes, the standards are analysed before and after the maps, as detailed in the discussion with reviewer 1. The standards are used for normalization of the Pb/Pb ratios, and drift correction of the U/Pb ratios, by bracketing. More details will be given in section 2.3. as also requested by the other reviewer.

RC2: "All figures with the robust regression have a white to blue, 0.x-1 colour scheme indicated on the right (I assume the weight as described in section 2.3.3?), but nowhere explained what it means. There are also open circle symbols likely the outliers that are not described. Please give this information either directly in the figures, or the figure caption, or leave it and just have points. Light blue to white points are hardly visible on white background."

The color scale has been changed. The key (weights) will be added. The problem of the open circle symbols no longer arises because they were linked to the rejection of outliers, which is no longer carried out.

RC2: "Section 7.1 Please give the sensitivity of your instrument as % of # ions detected of # of atoms ablated, e.g. for a volume measured crater in NIST 610 and a measurement of U only."

The percentage of ions detected with regard to atoms ablated is ca. 0.04% for U, as calculated with NIST 612 with the ICPMS method used for the dating map (comprising 6 isotopes).

RC2: "Please mention what kind of ablation cell is used, single volume, 2 volume, manufacturer, size, shape etc."

The laser ablation system is equipped with a home-made (home-designed) two volumes ablation cell. The large cell has a rectangular shape and a volume of 11.25 cm3 (75 x 25 x 6 mm size) while the small one, placed above the sample is of 10 mm diameter. These details will be added to the manuscript.

RC2: "What is the possible sample throughput of your system per day with the described method?"

The sample throughput is about 4 to 6 samples / day for the unknowns. It would certainly be higher with a more adapted equipment. We want to emphasize again that our work is aimed at highlighting the robust regression treatment, more than the devices used for the analyses.

RC2: "Out of curiosity, what kind of cones (Jet sampler and H or X skimmer) do you use in combination with the Jet Interface of your Element XR? (this does not need to be part of the manuscript)"

We use Ni-jet version with a Ni X-version skimmer.

RC2: "figure S2: mn should be min. Please indicate what is plotted either in the figure itself or the caption (238U/206Pb)?"

The requested modifications will be made.

―――――――――――――――――――――

---

## Author Response (AR1)

Guilhem Hoareau
Laboratoire des Fluides Complexes et leurs Réservoirs
IPRA-Université de Pau et des Pays de l'Adour
1 Avenue de l'Université
64013 PAU Cedex
France
Phone : +33 (0)5 59 40 73 40
guilhem.hoareau@univ-pau.fr

Pau, Thursday 24 September 2020

To *Geochronology* Associate Editor,

Dear Associate Editor,

Please find a new version of manuscript gchron-2020-10 "Direct U-Pb dating of carbonates from micron scale fsLA-ICPMS images using robust regression", by Hoareau et al.
As requested by the reviewers, the manuscript has been modified significantly.

- the text was largely rewritten, including dealing with the analytical strategy, to make it clearer;
- a systematic comparison between the robust regression approach and the one proposed by Drost et al. (2018), that we have reproduced with our own code, is now proposed for all samples;
- 2 samples presented in the study of Drost et al. (2018) (JS4 – run2 and BM18) have been also included in this perspective;
- we have added a new undated sample (C6-265-D5), for which the results differ significantly between the two approaches. Added to sample BM18, these differences are used to discuss the pros and cons of the robust regression approach;
- for the latter, we also introduce the use of weighted robust regression, which can help getting better results for low-U samples;
- the figures have been modified to include these modifications;
- a Table has been added, which presents all the ages and their uncertainties;
- as requested, we have added an electronic supplement with drift-corrected pixel values for the masses used in the study, to allow any interested geochronologist to "play with the data".

We hope that these modifications will be appreciated, and will allow this work to be published in the special issue of Geochronology.

Sincerely,

Guilhem Hoareau

[revised manuscript text omitted]
 (< 10), An age obtained from all the pixels can therefore potentially correspond in reality to a mixture of ages and distinct values of common lead. When taking the entire map to calculate an age, and some dispersion in the $^{207}Pb/^{206}Pb$ values for the weakly radiogenic pixels can be noticed (Fig. 6B7C). However, the U/Pb and Pb/Pb ratios of several pixels define a clear linear trend (Fig. 7C), allowing age calculation. The age obtained from the pseudo-spots is 100.2 ± 10.1 Ma (Fig. 7D, Table 1). Nevertheless, the agesThose obtained with the robust regression are identical, within the limits of uncertainty: 967.10 ± 3.94.6 Ma and 967.50 ± 4.08 for the TW (Fig. 6B7C) and 86TW plots (not shownFig. S3), respectively, and an age of 95.69 ± 3.35 Ma for the isochron plot (Fnot shownig. S3). These similar ages are consistent with the good statistical parameters obtained from the robust regression (MSWD-d < 1.2, RSE lower than 8.5% of the $Pb_0$ value). However, both the maps of pixel concentration values and the colocalization analysis of image data demonstrate that the overdispersion of $^{207}Pb/^{206}Pb$ values are located in a restricted area of the map corresponding to one cement (cement C1 on the CL image, Fig. 6C7B), whereas the highest $^{238}U/^{206}Pb$ (i.e., most radiogenic values) that drive the age are located in another restricted area corresponding to another cement (cement C2 on the CL image) (Fig. 6D7E). Lead by the colocalization study, the selection of a small area of the map corresponding to cement C2 results in similar but more accurate age calculations than considering the entire map (97.46 ± 34.52 Ma for the TW plot (Fig. 6E7F), 967.91 ± 34.51 Ma for the 86TW (Fig. S3) and 95.54 ± 3.4 3 Ma for the isochron plot (Fig. S3)). The statistical parameters are also better, despite a number of pixels divided by three (MSWD-d < 0.9, RSE lower than 5% of the $Pb_0$ value).

[revised manuscript text omitted]

---

## Author Response (AR2)

Guilhem Hoareau
Laboratoire des Fluides Complexes et leurs Réservoirs
IPRA-Université de Pau et des Pays de l'Adour
1 Avenue de l'Université
64013 PAU Cedex
France
Phone : +33 (0)5 59 40 73 40
guilhem.hoareau@univ-pau.fr

Pau, Friday 27 November 2020

To *Geochronology* Associate Editor,

Dear Associate Editor,

Please find a final, revised version of manuscript gchron-2020-10 "Direct U-Pb dating of carbonates from micron scale fsLA-ICPMS images using robust regression", by Hoareau et al.

All the last, minor modifications requested by the 2 reviewers have been done in order to publish the manuscript:

**#Reviewer 1:**

*Line 18: If the ICP-MS is not used in high resolution mode I suggest calling it sector filed ICP-MS instead of high resolution ICP-MS.*

Done

*Line 44: Replace "standard" by reference material. (throughout the manuscript). Only material certified by an institution should be called standard, all other materials we use for correction and validation should be called reference material (RM) and validation RM.*

Done

*Line 200-201: Why are the systematic uncertainties given as 1S, when all the results are given as "1.96σ level" ? That is a bit confusing.*

Modified

*Where is the long-term excess variance of 1% (S1) from? (Guillong et al 2020) suggest 2-2.5% (2S), please comment on how you got 1%.*

An explanation is now given directly in the manuscript (l. 200-204).

**#Reviewer 2**

*page 6/lines 170-2*

*Whether or not a drift correction is applied is still not really clear as the inserted sentence represents a contradiction ("correction for drift was also based on a standard bracketing approach" vs. "using the values obtained for the NIST612 preceding the unknown")*

The sentence has been modified and made clearer (l. 170-171)

*page 11/ sample JS4*
*The information given on sample JS4 from Drost et al. (2018) is inconsistent. Fig. 6A shows n=980, which is consistent with the data set given in Fig. 3A-D of Drost et al. (2018) (JS-4 run 1, criteria: Rb, Zr, Th<0.3 ppm, obtained ages TW 300.8±3.7Ma, 86TW 299.6±4.0Ma, isochron 298.9±5.6Ma). However, on page 4/line 97 you quote 300.5±3.3Ma (which belongs to run2 in the original publication), on page 11/line 333 you quote run 2 in the header and in lines 335-6 you give the ages for run 1 with criteria: Rb<0.5ppm. Please check and clearify.*

Corrected

*page 12/line 343*
*"Like JS4, .....is characterized by low U and Pb...." should rather read "In contrast to JS4, .... is characterized by low U and Pb...."*

We meant "Unlike": corrected

*page 13/line 399*
*"... U/Pb and Pb/Pb ratios (~0.2 to ~106 for 238U/206Pb),..." - 238U/206Pb is rather around 70 in Fig. 8B*

The numbers are the right ones, but a few pixels are absent from the figure due to the scale chosen. This is now clearly stated in the caption (also for Figs. 6 and 9)

*page 14/lines 403 and 406*
*ages reported here are not the same as in Fig. 8D and 8E, respectively*

Corrected

*page 14/line 415*
*reference to Fig. 8C is not correct - should be 8E instead?*

Corrected

*pages 14/15*
*numbering of sections is still inconsistent (section 3.2.3. is followed by 5.)*

Corrected

*7.1 Analytical conditions table*
*Quality control validation: "4 analyses of Duff Brown..." but then there are five ages given*

Corrected

*Table 1*

*here and in the figures and supplementary figures: the rounding is inconsistent*

Corrected

*Figure captions:*
*Fig. 5:*
*"D: TW concordia plot obtained from in situ LA-ICPMS analyses.." Do you want to replace "in situ" by "spot"?*

Done

*Inconsistent numbering in the captions (instead of 1, 2, 3....9 it is 1, ...5, 6, 6, 7, 8). Numbering in the text seems to be correct though.*

Corrected

*Supplementary material (pdf)*
*you may want to consider replacing "Pb0 fixed" and "Pb0 not fixed" by "anchored to Pb0" and "unanchored"*

Done

*Duff Brown 1 in the supplementary material is called Duff Brown 2 in Table 1*

Corrected (in the supplementary)

*Supplementary Table S1 (xlsx)*
*"This file presents the isotopic images, in number of counts, for the masses 238, 232, 208, 207, 206, for samples Duff Brown, BH14, PXG20-1, PXG32-2, ETC2, ARB and C6-265-D5 ."*
*- should it read "counts per second" instead of "number of counts"?*

Yes. Corrected.

Sincerely,

Guilhem Hoareau

[revised manuscript text omitted]

---

## Author Response (AR3)

Guilhem Hoareau
Laboratoire des Fluides Complexes et leurs Réservoirs
IPRA-Université de Pau et des Pays de l'Adour
1 Avenue de l'Université
64013 PAU Cedex
France
Phone : +33 (0)5 59 40 73 40
guilhem.hoareau@univ-pau.fr

Pau, Friday 18 December 2020

To *Geochronology* Editor,

Dear Editor,

Please find a final, revised version of manuscript gchron-2020-10 "Direct U-Pb dating of carbonates from micron scale fsLA-ICPMS images using robust regression", by Hoareau et al.

The last, minor modifications you requested have been done in order to publish the manuscript.

Rather than strictly following the adequate number of significant digits, we have homogenized the number of decimal places between the results and their uncertainty, without ever exceeding 3 decimal places. Note that these modifications were made on all the figures in the manuscript, but in the supplement only the figures generated by our own code (robust regression) were modified. Indeed, for the figures obtained via IsoplotR, we have kept the default presentation proposed by the software (i.e., 2 significant digits). On this last point, we notice that the 2 significant digits are calculated by default only for uncertainties, which may result in aberrations in the number of significant digits for the results. This problem is therefore encountered by all users of the software.
Finally, we have homogenized the format of the common Pb in the figures (for example, $^{207}Pb/^{206}Pb)_0$ rather than $Pb_0$), following your recommendations but also the default format proposed in IsoplotR.

We hope that these modifications are satisfactory to you and will allow the rapid publication of our results.

Sincerely,

Guilhem Hoareau

[revised manuscript text omitted]